# Density Induced Vacuum Instability

Reuven Balkin[1,2], Javi Serra[1], Konstantin Springmann[1]*, Stefan Stelzl[1] and Andreas Weiler [1]

**1** Physik-Department, Technische Universität München, 85748 Garching, Germany
**2** Physics Department, Technion – Israel Institute of Technology, Haifa 3200003, Israel
* konstantin.springmann@tum.de

November 25, 2022

## Abstract

We consider matter density effects in theories with a false ground state. Large and dense systems, such as stars, can destabilize a metastable minimum and allow for the formation of bubbles of the true minimum. We derive the conditions under which these bubbles form, as well as the conditions under which they either remain confined to the dense region or escape to infinity. The latter case leads to a phase transition in the universe at star formation. We explore the phenomenological consequences of such seeded phase transitions.

# 1  Introduction

The rich and versatile physics of light scalar fields is behind their central role in many scenarios that address the shortcomings of the standard models of cosmology and particle physics, such as the nature of dark matter and dark energy, or the electroweak hierarchy and strong-CP problems. A particularly interesting aspect of scalar dynamics is associated with the presence of multiple minima of the scalar potential, leading to a plethora of phenomena like false vacuum decay [1,2], early universe phase transitions [3,4], or vacuum selection of a small cosmological constant [5–7] or a small electroweak scale [8–12]. Similar to the well-known case of finite temperature, in which the coupling of the scalar field to a thermal bath changes the structure of its potential and opens the door to transitions between different minima, in this work we wish to explore the much less studied question of the fate of an in-vacuo metastable ground state at finite density.

Finite density effects on scalar potentials have long been considered for the QCD order parameters, see e.g. [13,14], as well as in the context of chameleon field theories, see [15] for a review. Moreover, it has been recently shown that the potential of the QCD axion [16], and of certain deformations thereof [17], changes in systems with large baryonic densities, such as neutron stars. In these examples, the coupling of the scalar to a background matter density (either total or a specific subcomponent), can displace the field away from its value in vacuum. Here we go one step further and investigate how finite density effects on scalar potentials with multiple minima can give rise to field displacements large enough to reach the value of a *lower energy* minimum. This possibility could be realized in e.g. relaxion models [10], where the scalar potential is a tilted cosine with its magnitude set by the QCD quark condensate or the Higgs VEV, which are sensitive to densities as those found in stars.

To make the discussion of the physics as transparent as possible, in this paper we work with a simple potential à la Coleman [1], that is a quartic function of a single scalar field $\phi$, with a $Z_2$ symmetry $\phi \to -\phi$, which is explicitly broken by a linear term, and with the scalar field in vacuum sitting at the metastable minimum. The barrier separating the two minima is argued to decrease with density, thus for sufficiently high densities the metastable minimum disappears, leading to the formation of a non-trivial scalar profile within the dense system (which for simplicity we model as a spherically symmetric compact object, i.e. a star). Inside this scalar bubble, the field is displaced from its position in vacuum and, if the system is large enough, it acquires a value that corresponds to the true minimum of the potential.[1] Interestingly, we find that depending on the density profile and evolution of the star, an instability takes place such that the bubble permeates through the entire system, escapes and propagates to infinity, on account of the fact that the scalar inside the bubble is in the preferred energy configuration also in vacuum.

These seeded phase transitions could have catastrophic implications for our universe. Since our main focus is on transitions to the true vacuum that are classically allowed, they take place as soon as stars that are dense and large enough are formed. Such a late phase transition, at redshifts no earlier than $z \sim 20$, changes the vacuum energy with respect to that inferred from measurements of the CMB. This allows us to place bounds on the parameters of the scalar

---

[1]A similar situation has been previously considered in [18], yet there the scalar field in vacuum lies at the true minimum, therefore the scalar bubble remains confined within or around the star.

potential that depend on the type of stars triggering the phase transition.[2] Still, if the energy difference between the two minima is sufficiently small, such phase transitions could be non-lethal and potentially detectable with future cosmological and astrophysical observations.

The rest of the paper is organized as follows. In Sec. 2 we present the scalar potential we take as case study and discuss how it can change at finite density. Sec. 3 is devoted to the description of the essential properties of the systems of interest, i.e. the stars. Classical bubble formation and dynamical evolution are discussed in Sec. 4, along with the derivation of the conditions leading to bubble escape. In this section we also comment on quantum bubble formation via tunneling assisted by finite density. In Sec. 5 we explore the main phenomenological consequences of a late-time phase transition and derive the corresponding constraints on the scalar potential. Finally, we present our conclusions and outlook in Sec. 6. In several appendices we discuss some supplementary approximations and the relevance of ultra-high densities as well of gravitational forces on the bubble dynamics.

## 2   Scalar potential

The potential we consider is just the familiar quartic potential with a linear tilt,

$$V(\phi) = -\frac{1}{3\sqrt{3}} \Lambda_{\text{R}}^4 \frac{\phi}{f} + \frac{1}{8} \Lambda_{\text{B}}^4 \left( \frac{\phi^2}{f^2} - 1 \right)^2 . \tag{1}$$

$\Lambda_{\text{R}}$ and $\Lambda_{\text{B}}$ are the scales that control the size of what we denote as linear "rolling" and quartic "barrier" terms respectively (numerical factors are introduced for notational convenience), while $f$ parametrizes the field distance between the two minima. The potential has two minima as long as

$$\delta^2 \equiv 1 - \frac{\Lambda_{\text{R}}^4}{\Lambda_{\text{B}}^4} > 0 . \tag{2}$$

For $1 - \delta^2 \ll 1$ the minima are located at $\phi_\pm \simeq \pm f$, and in particular the metastable minimum $\phi_-$ is a *deep* minimum. Instead, for $\delta^2 \ll 1$ the minima are at $\phi_- \simeq -f/\sqrt{3}$ and $\phi_+ \simeq 2f/\sqrt{3}$, and $\phi_-$ is *shallow*. The difference between these two types of metastable minima is evident from the mass of the scalar

$$m_\phi^2 \simeq \begin{cases} \sqrt{\frac{2}{3}} \frac{\Lambda_{\text{B}}^4}{f^2} \delta , & \text{(shallow)} \\ \frac{\Lambda_{\text{B}}^4}{f^2} . & \text{(deep)} \end{cases} \tag{3}$$

For a shallow minimum ($\delta^2 \ll 1$) the mass is parametrically suppressed with respect to the usual expectation, which is instead reproduced in the case of a deep minimum ($1 - \delta^2 \ll 1$). Another quantity of phenomenological interest, which is markedly different between shallow and deep minima, is the height of the potential barrier,

$$\Delta V_{\text{top}} \simeq \begin{cases} \frac{4}{27} \sqrt{\frac{2}{3}} \Lambda_{\text{B}}^4 \delta^3 , & \text{(shallow)} \\ \frac{1}{8} \Lambda_{\text{B}}^4 . & \text{(deep)} \end{cases} \tag{4}$$

The suppression of the barrier in the case of minima with $\delta^2 \ll 1$ implies that even a small perturbation of the potential can easily destabilize the scalar field.

Let us note that while shallow metastable minima might naively be deemed as tuned, they naturally appear in relaxion models [10], where the barrier term is a periodic function of the

---

[2]The implications of these findings for relaxion models are presented in [19]. Part of these results have been advanced in [20] and later in [21,22].

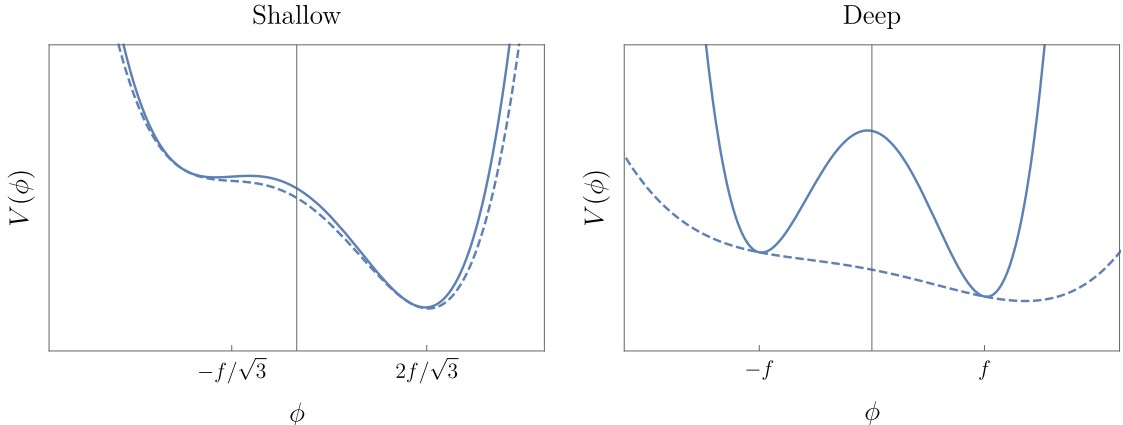

Figure 1: Potentials with shallow (left) and deep (right) minima in vacuum (solid) and in medium for a density $n$ slightly larger than critical (dashed).

scalar field, e.g. $\cos(\phi/f)$, whose amplitude increases very slowly with each $\phi$ oscillation. There, the first minima of the potential are found when the barriers get just large enough, i.e. $\Lambda_{\text{B}}^4 \approx \Lambda_{\text{R}}^4$, or in our notation $\delta^2 \ll 1$. The quartic potential we have taken as a case study in Eq. (1) is a simplified version of the relaxion case.

## 2.1 Finite density

Finite density can impact a scalar potential in several ways, depending on how the scalar couples to the matter fields that constitute the dense system. In general, density corrections can be encoded as an additional term in the potential that explicitly depends on density, $n$, and vanishes in vacuum, i.e. $n = 0$. For the sake of concreteness, in this work we focus on the scenario where these corrections can be entirely encoded as a non-trivial density dependence of the parameters of the potential Eq. (1). In particular, we consider the situation where the barrier $\Lambda_{\text{B}}$ depends on density, and define the dimensionless quantity

$$\frac{\Lambda_{\text{B}}^4(n)}{\Lambda_{\text{B}}^4} \equiv 1 - \zeta(n), \tag{5}$$

with $\zeta(n) \geqslant 0$ and $\zeta(0) = 0$.

This scenario is naturally realized when $\Lambda_{\text{B}}$ itself is determined by the vacuum expectation value of an operator that is sensitive to finite density corrections. Perhaps the simplest example in the SM is provided by the QCD quark condensate, that is $\Lambda_{\text{B}}^4 \propto \langle \bar{q}q \rangle \sim \Lambda_{\text{QCD}}^3$, which is well-known to linearly decrease with (small) baryon density $n_b = \langle B^\dagger B \rangle$ [13]. In the notation of Eq. (5), this would imply, at leading order in density, that $\zeta(n_b) \propto n_b/\Lambda_{\text{QCD}}^4$ in systems with a non-zero nuclear density, such as stars. The case of a $\Lambda_{\text{B}}$ proportional to any other QCD condensate that is non-zero in vacuum and changes with baryon density, such as a gluon condensate, belongs to the same class. Within the realm of SM operators, the only other qualitatively different case is given by a barrier set by the Higgs VEV, that is $\Lambda_{\text{B}}^4 \propto \langle h^2 \rangle = v^2$. There, the coupling of the Higgs field to fermions, $y_\psi h \bar{\psi} \psi$, displaces its expectation value when in a (non-relativistic) $\psi$ background, $\langle \bar{\psi} \psi \rangle \simeq \langle \psi^\dagger \psi \rangle \neq 0$. Considering once again a system with a non-vanishing baryon density, the small displacement in the Higgs would lead, at leading order, to $\zeta(n_b) \propto n_b/m_h^2 v^2$. Let us note that $\Lambda_{\text{B}}^4 \propto \langle \bar{q}q \rangle$ is realized by the QCD-axion [16,17], as well as by those models of relaxation of the electroweak scale where the relaxion is identified with the QCD-axion [10]. The case where the leading finite density effects are due to a shift of the Higgs field, $\Lambda_{\text{B}}^4 \propto \langle h \rangle^2$, is found in non-QCD relaxion models [10], and

it could arise as well in more general Higgs-portal models, e.g. [23]. A detailed discussion of finite density effects in these versions of the relaxion can be found in [19]. Going beyond the SM, we could, for instance, entertain the possibility that $\Lambda_{\text{B}}$ originates from the confinement of a new QCD-like dynamics decoupled from the SM. Motivated by this case, we should further consider the existence of dark compact objects, a.k.a. dark stars [24–32], whose non-zero density can lead to a change of the scalar potential as in Eq. (5).

Because of the smaller barriers at finite density, the metastable minimum in vacuum is no longer a minimum in a dense system as soon as the condition Eq. (2), with $\Lambda_{\text{B}}^4 \to \Lambda_{\text{B}}^4(1-\zeta)$, is not satisfied. The critical value of $\zeta$ above which this destabilization occurs is

$$\zeta_c = 1 - \frac{\Lambda_{\text{R}}^4}{\Lambda_{\text{B}}^4} = \delta^2 \,. \tag{6}$$

It is evident from this expression that a shallow local minimum is more easily destabilized than a deep one, since $\zeta_c \ll 1$ for a shallow minimum while $\zeta_c \approx 1$ for a deep one. This is explicitly shown in Fig. 1. For reasonable scenarios where $\zeta(n)$ increases with $n$, the critical density $n_c$, defined by $\zeta(n_c) = \zeta_c$, required for the local minimum to disappear is much lower for a shallow than for a deep minimum. We limit our discussion to $\zeta(n) \leqslant 1$, since otherwise the barrier term changes sign and the scalar potential is no longer bounded from below. This makes the analysis sensitive to higher-order terms in $\phi$, which we have implicitly neglected; in other words, the scalar dynamics becomes UV sensitive and therefore no longer predictive. In addition, note that for what concerns the destabilization of the false vacuum, the relevant quantity is the ratio between the rolling and barrier scales. Therefore, we could just as well have considered a density dependent rolling term, $\Lambda_{\text{R}}^4(n)$, as the source of the instability. However, as we show in Sec. 4, the formation of a scalar bubble within a dense system of finite size, as well as its evolution, mostly depends on the magnitude of the rolling term. For this reason, in this work we keep $\Lambda_{\text{R}}$ density independent. Let us also point out that density is treated here a background field that eventually depends on space and time, see Sec. 3. Although we are phrasing our discussion of the fate of the metastable minimum in terms of a matter density, a priori other space-time dependent background fields could lead to similar effects on the scalar potential. An example where the rolling scale is modified is presented in [19], where the role of density is played by a background electromagnetic field.

As discussed above, for densities above the critical one, the scalar potential has a single minimum. We denote this minimum as $(\phi_+)_n$, such that it is clear that it is continuously connected, as the density is taken to zero, to the stable minimum in vacuum, $\phi_+$. Let us note that close to criticality, i.e. for $\zeta(n) \simeq \zeta_c$, the in-density potential has the same form as a potential in vacuum with $\delta^2 \ll 1$, thus $\phi_{+\, n \simeq n_c} \simeq 2f/\sqrt{3}$. For the same reason, just before the critical density is reached, the in-medium metastable minimum is shallow and found at $-f/\sqrt{3}$, regardless of its value in vacuum $\phi_-$. In contrast, far beyond the critical density, the single minimum of the potential is found at

$$\phi_{+\, n \gg n_c} \sim \left( \frac{1-\zeta_c}{1-\zeta(n)} \right)^{1/3} f \,, \tag{7}$$

which can be much larger than $f$ if $\zeta \to 1$. Whenever the scalar potential has two minima, be these shallow or deep, at zero or non-zero density (obviously as long as $n < n_c$), the difference in the ground state energy between them is given by

$$\Delta\Lambda \sim -\Lambda_{\text{R}}^4 \,, \tag{8}$$

up to an irrelevant $O(1)$ factor.

We would like to emphasize that while in this work we focus on a simple potential of the form Eq. (1), the analysis presented in this section as well as subsequent sections can be

applied as well to other types of potentials containing local minima separated by a density-dependent barrier. Furthermore, even though we pay particular attention to the fact that at finite density the scalar field can classically move to the true minimum of the potential, this is not the only case of interest; such a change of minimum could be classically forbidden at finite density as well, yet take place anyway due to a much shorter quantum-mechanical lifetime than in vacuum (see Sec. 4.4).

Furthermore, a comment is in order regarding the UV sensitivity of the scalar potential Eq. (1) and our assumptions on how it changes at finite density. Indeed, let us consider the case that $\Lambda_{\text{B}}^4 = \alpha \langle h \rangle^2$, where $\alpha$ is just a proportionality factor. By closing the Higgs loop and cutting it off at a scale $\Lambda_h$, we obtain a contribution to the barrier term $\Delta \Lambda_{\text{B}}^4 \sim \alpha (\Lambda_h / 4\pi)^2$. We should then demand that this extra contribution does not erase the instability of the local minimum at finite density, which means $\Delta \Lambda_{\text{B}}^4 \ll \Lambda_{\text{B}}^4(n_c) \simeq \Lambda_{\text{R}}^4$. This conditions translates into an upper bound on the cutoff of the scalar theory, $\Lambda_h \ll 4\pi \langle h \rangle \sqrt{1 - \delta^2}$. Note this is larger for potentials with a shallow metastable minimum than for those with a deep minimum. Such a low cutoff does not endanger our analysis of the scalar field dynamics at finite density as long as $\Lambda_h \gg E_{\text{S}}$, where $E_{\text{S}}$ is the typical energy scale of the dense system. Similar conclusions apply to the other possible cases concerning the density dependence of $\Lambda_{\text{B}}$, see the discussion below Eq. (5).[3] Besides, already from the quartic scalar interaction in Eq. (1), naturalness arguments indicate that new physics should appear at a scale $\Lambda_\phi \sim 4\pi f$ or below. Once again, we should demand that $\Lambda_\phi$ is significantly above $E_{\text{S}}$.

Finally, throughout this work we will be agnostic about the details of the cosmological history which lead to the initial condition under consideration, i.e. a universe at a meta-stable minimum at late cosmological times. While this metastable minimum could a priori be destabilized in the early universe, it is also easy to devise a scenario in which it survives until later times. Consider, for example, the possibility of vacuum destabilization due to finite temperature effects. This would depend on the temperature necessary to eliminate the potential barrier, e.g. $T \sim 100\,\text{MeV}$ for a QCD barrier $\Lambda_{\text{B}}^4 \propto \langle \bar{q}q \rangle$, $T \sim 100\,\text{GeV}$ if $\Lambda_{\text{B}}^4 \propto \langle h \rangle^2$, or even the deconfinement temperature of the new QCD-like sector that generates $\Lambda_B$, if this is thermalized. However, it could well be that such high enough temperatures were never reached in the early universe. In this regard, not much is known about the history of the universe above the temperatures of Big Bang Nucleosynthesis (BBN), $T_{\text{BBN}} \sim 10\,\text{MeV}$, where in addition baryonic densities were as low as $n_{\text{BBN}} \sim 10^{-10}\,\text{MeV}^3$, see e.g. [17]. More generally, we would like to stress that we can think about our two minimum potential as part of a broader landscape of vacua. As an example, for relaxion models it has been shown that due to the overall flatness of the relaxion potential, even if reheating temperatures were high enough as for the barriers between minima to disappear, the universe would have cooled so fast that the relaxion would not have rolled down many minima of its potential [10, 33]. This means that the relaxion would have eventually landed in some minimum that is susceptible to being destabilized by stellar compact objects.

# 3   Spherically symmetric dense systems

In this work we are interested in dense systems of finite size, in particular stars. We model the star as a spherically symmetric (non-rotating) object with a density profile that in general

---

[3]$\Lambda_{\text{B}}$ is insensitive to the UV if e.g. the barrier term arises from the coupling of the scalar to the QCD topological charge, i.e. $\frac{1}{f}\phi G\tilde{G}$, which gives rise to a potential sensitive to $\Lambda_{\text{QCD}}$ only. For instance, this is the case of the QCD-relaxion, where we recall that the corresponding scalar potential is of the form $\cos(\phi/f)$ instead of the simple quartic function we are considering.

depends on radius and time, i.e. $n(r, t)$. The profile satisfies ($n' = dn/dr$),

$$n'(0, t) = 0, \qquad n(R_{\text{s}}(t), t) = 0, \tag{9}$$

such that the density profile is differentiable at the origin, $r = 0$, and that the star ends at a finite radius, $r = R_{\text{s}}$, respectively. In addition, we define a transition radius, $r = R_{\text{T}}$, where the critical density is reached,

$$n(R_{\text{T}}(t), t) = n_c. \tag{10}$$

We recall that at densities above critical, the local minimum of the potential is lost.

Since the scalar potential at finite density is minimized at a different value than in vacuum, minimization of the action forces the field to acquire a (spherically symmetric) non-trivial profile within and around the star, $\phi(r, t)$. This is determined by the classical EOM ($\dot\phi = d\phi/dt$, $\phi' = d\phi/dr$ and $V_{,\phi} = dV/d\phi$)

$$\ddot\phi - \phi'' - \frac{2}{r}\phi' = -V_{,\phi}, \tag{11}$$

where $V = V(\phi, n(r, t))$, with the boundary conditions

$$\phi'(0, t) = 0, \qquad \lim_{r \to \infty} \phi = \phi_-. \tag{12}$$

In order to solve Eq. (11) one needs to know the density profile of the star, which generically depends on non-trivial and in some cases not well-understood dynamics (e.g. the inner regions of neutron stars). If there is a large separation of scales in the problem, we can, as a first approximation, be agnostic of the details of the density profile, as we explain in the following. The characteristic scale controlling the classical evolution of the scalar profile, either in time or space, is determined by its potential. For the representative case that we are considering, Eq. (1), the EOM for the dimensionless field $\hat\phi \equiv \phi/f$ can be written as

$$\frac{\partial^2 \hat\phi}{\partial \hat t^2} - \frac{\partial^2 \hat\phi}{\partial \hat r^2} - \frac{2}{\hat r}\frac{\partial \hat\phi}{\partial \hat r} = 1 - \frac{3\sqrt{3}}{2}\frac{1-\zeta}{1-\zeta_c}(\hat\phi^2 - 1)\hat\phi, \tag{13}$$

where $\hat r = \mu r$, $\hat t = \mu t$, and

$$\mu^2 = \frac{1}{3\sqrt{3}}\frac{\Lambda_{\text{R}}^4}{f^2} \sim \frac{\Lambda_{\text{R}}^4}{f^2}. \tag{14}$$

For densities sufficiently above the critical one, such that $1 - \zeta \ll 1 - \zeta_c$, $\mu^{-1}$ sets the typical time and distance required for the scalar to move by $\Delta\hat\phi = O(1)$. This is to be compared with the characteristic scales of the dense system.

Let us first discuss time evolution, i.e. the formation of the star. The dimensionless quantity $\mu T_{\text{s}}$, where $T_{\text{s}}$ is the characteristic time scale of the dense system, gives us a rough idea whether we can treat the evolution of the scalar field as effectively taking place in a nearly static, fixed system, or whether the time dependence of the scalar profile is comparable to (or much slower than) the typical time scale of the star. Indeed, for $\mu T_{\text{s}} \gg 1$ the field reacts fast to changes in the background density profile, therefore we can describe the scalar dynamics as a *quasi-static* (or *adiabatic*) process, in which $\dot\phi$ and additional time derivatives can be neglected. On the other hand, for $\mu T_{\text{s}} \ll 1$ the field reacts slow compared to the evolution of the star, in which case the evolution of the scalar profile can be described in a *sudden* (or *non-adiabatic*) approximation, where the formation of the star can be treated as an instantaneous change from vacuum to $n(r) \neq 0$ and $\phi$ starts "rolling" down the in-medium potential.

In the adiabatic limit, $\mu T_{\text{s}} \gg 1$, the scalar profile can be found at any given time $t = \bar t$ during the formation of the star by solving its *time-independent* EOM, within a fixed background

density $n(r) = n(r, \bar{t})$.[4] We shall consider simple density profiles that can be parametrized as

$$n(r) = n_o(\bar{t}) \, g(r/R_s(\bar{t})), \tag{15}$$

where the function $g(x)$ fully encodes the radial dependence, with $g(0) = 1$ such that the density at the center is set by $n_o$, $g(R_T/R_S) = n_c/n_o$, and $g(1) = 0$. While obtaining the specific form of $n(r)$ at a given $\bar{t}$ is generically a complicated problem, the only quantities of qualitative relevance for our analysis are $R_T$, the radius below which the critical density is surpassed, that is where the in-vacuo potential barrier disappears and the scalar can potentially be displaced by $O(f)$, and $\Delta R_T = R_S - R_T$, the size of the transition region towards the end of the star, where the potential barrier reappears. We find that non-trivial dynamics take place when $\mu R_T \sim 1$, and additionally when $\mu \Delta R_T \sim 1$, see Sec. 4. The value of $R_T$ depends on the value of the critical density, which in turn depends on how the scalar potential changes with density. For typical density profiles in which the central density is significantly larger than the critical one, one generically finds $R_T \sim R_S$ [34, 35]. This then implies that $\Delta R_T \sim R_S$ as well. In addition, since in practice each class of stars, e.g. neutron stars, white dwarfs, or main-sequence stars like the Sun, covers a range of radii, we also expect to find a range of values for $R_T/R_S$ and $\Delta R_T/R_S$, where generically both ratios are $O(1)$.

In this paper we concentrate on the adiabatic limit just described. Since a non-trivial scalar profile develops when $\mu R_S \sim 1$, we focus on stellar processes where the relevant time scale is $T_S \gg R_S$. As an example, let us discuss the interesting case of neutron stars, since they exhibit the largest (baryonic) densities and the fastest dynamics, and assume that densities prior to the birth of the neutron star are below the critical density, which to be concrete we fix at nuclear saturation density, $n_c = n_0 \approx 0.16/\text{fm}^3 \approx (110\,\text{MeV})^3$. The birth of a neutron star follows from the gravitational collapse of the core of a massive star, which leads to a supernova (SN) explosion, see e.g. [36, 37]. While the details of this process are not completely understood, it has been reliably inferred that densities reach and surpass nuclear saturation in a time $T_S = T_{NS} \sim 1\,\text{s}$. Within this time, the size of the core of the star in which densities have exceeded $n_0$ is an $O(1)$ fraction of the total size of the final neutron star, i.e. $R_T \sim R_S = R_{NS}$. Since the typical radius of a neutron star is $R_{NS} \sim 10\,\text{km}$, we find $R_{NS} \ll T_S$, justifying the quasi-static approximation. Similar conclusions can be reached for other types of stars, for instance white dwarfs, with typical radii $R_{WD} \sim 10^3\,\text{km}$ and densities $n_{WD} \sim \text{MeV}^3$, or the Sun $(R_\odot \approx 7 \times 10^5\,\text{km}, n_\odot \approx 7 \times 10^{-9}\,\text{MeV}^3)$. Note that the above-mentioned densities set the typical scales in the potential at which our mechanism is relevant. In any case, for completeness we briefly discuss the regime $\mu T_S \ll 1$ in App. E.

For the reader's reference, the scale $\mu^{-1}$ is of order of the typical size of a neutron star for e.g. the potential parameters

$$\mu R_S \sim 5 \left( \frac{R_S}{10\,\text{km}} \right) \left( \frac{\Lambda_R}{10\,\text{eV}} \right)^2 \left( \frac{1\,\text{TeV}}{f} \right). \tag{16}$$

Several additional comments are in order. First, in the special case that the (central) density happens to be very close to $n_c$, one naturally expects $R_T \ll R_S$, making the analysis more sensitive to the specifics of the density profile. Second, since the reaction time of the scalar gets suppressed by $\zeta - \zeta_c$, the adiabatic approximation naively fails at some arbitrarily small time interval around the time in which $\zeta \to \zeta_c$.[5] Lastly, our study neglects the effects of temperature altogether. This is a good approximation in most situations, yet for e.g. the Sun as

---

[4]In practice, numerically calculating these static bounce-like solutions is challenging since it requires a finely-tuned boundary condition at the origin. More details on our numerical calculations can be found at the end of this section.

[5]In a spatially homogeneous situation, the local condition that determines if the scalar is able to follow the minimum of the potential can be expressed as $\dot{\phi} \gtrsim (d\phi_{\min}/dn)\dot{n}$, where $\phi_{\min} \simeq \phi_-$ as long the metastable minimum exists, and $(\phi_+)_n$ otherwise. However, if the system exhibits a non-trivial spatial dependence, that this condition is

well as in SN explosions, temperature could be as important as density, i.e. $T^3 \sim n$. Neverthe-less, we note that for the motivated cases in which $\Lambda_{\text{B}}^4 \sim \Lambda_{\text{QCD}}^3$ or $\Lambda_{\text{B}}^4 \sim v^2$, the effect of a finite temperature would generically go in the same destabilizing direction as density, i.e. decreasing the size of the potential barriers, reinforcing our conclusions regarding the formation and escape of a scalar bubble.

Let us conclude this section by briefly discussing our numerical analysis. In order to verify the theoretical results we present in Sec. 4, we have solved the *time-dependent* EOM presented in Eq. (13) numerically, assuming simple dependencies, e.g. $\zeta \propto n(r, t)$. The initial conditions for the scalar field are homogenous, i.e. $\phi(r, 0) = \phi_-$ and $\dot{\phi}(r, 0) = 0$. We implement a slow evolution of the density profile from $n(r, 0) = 0$ to some final configuration Eq. (15) at $\bar{t} = T_{\text{s}}$, with $g(x) = 1 - x^2$. Importantly, we fix $\mu T_{\text{s}} \gg 1$, in agreement with the adiabatic limit. We verify that the quasi-static solutions we find have negligible amounts of kinetic energy compared to their gradient and potential energies. This quasi-static picture is maintained up until an instability takes place, i.e. until our numerical simulations display an expanding bubble that escapes from the star. Importantly, under our assumptions, the exact details of the star formation do not affect the quantitative scaling we present in the next section for the formation and escape of scalar bubbles.

## 4 Bubble formation and evolution

The formation of a non-trivial scalar profile induced by a star is effectively described, as justi-fied in Sec. 3, by the quasi-static spherically-symmetric EOM for the scalar field, with a slowly-varying background density profile. The bubble-like solution $\phi(r)$ can be found numerically given a specific form for the density profile $n(r)$. The simple analytic results presented in this section have been explicitly verified by our numerical simulations.

A few simplifications allow us to analytically understand the dynamics of scalar bubbles at finite density. The field profile minimizes the total energy,

$$E(R) \simeq 4\pi \int_0^R \mathrm{d}r\, r^2 \left[ \frac{1}{2}\phi'^2 + \Delta V(\phi, n) \right], \quad \Delta V(\phi, n) = V(\phi, n) - V(\phi_-, n), \qquad (17)$$

where we have cut the integral at a radius $R$ as an approximation to the full infinite space, since the scalar field rapidly converges to its vacuum value $\phi_-$ for $r \gtrsim R$. Indeed, for radii larger than the transition radius, i.e. $r > R_{\text{T}}$, densities are below critical and the potential is minimized at approximately the same metastable minimum as outside of the star. In the initial stages of the formation of the dense system, we expect the creation of a scalar *proto-bubble* with $R \simeq R_{\text{T}}$, where the scalar field at its center, $\phi(0)$, has not yet reached $\phi_+$, the value associated with the stable minimum of the in-vacuum potential, see Sec. 4.1. In other words, the field displacement, $\Delta\phi(0) \equiv \phi(0) - \phi_-$, satisfies $\Delta\phi(0) \lesssim \phi_+ - \phi_- \approx 2f$. This is because the star is too small, in particular the (mean) energy density in the field gradient that would correspond to a field displacement $\Delta\phi(0) \sim 2f$, which is $\frac{1}{2}\langle\phi'^2\rangle \sim (2f/R_{\text{T}})^2$, is too large compared to the (mean) potential energy difference within the proto-bubble, $\epsilon = |\langle\Delta V\rangle|$. Only when the star, by which we mean $R_{\text{T}}$, grows large enough, it becomes energetically favorable to reach

satisfied does not imply that the scalar actually follows the minimum; it becomes crucial to consider the gradient energy of the field, which impedes large field displacements. Besides, we note that right at the critical point, $n = n_c$, there is a discontinuous jump in $\phi_{\text{min}}$ (from $\phi_-$ to $(\phi_+)_n$) and therefore $d\phi_{\text{min}}/dn \to \infty$; equivalently, at the critical density $\mu \to 0$, since $\zeta = \zeta_c$, thus $\mu T_{\text{S}} \to 0$. Time dependence can then become important, yet only if the system is large enough to render the gradient energy negligible compared to kinetic energy the field acquires rolling down the potential.

$\phi(0) \sim \phi_+$. Therefore, only when

$$\left(\frac{2f}{R_{\mathrm{T}}}\right)^2 \lesssim \epsilon \tag{18}$$

can a scalar bubble *fully form*. Interestingly, once the condition Eq. (18) is satisfied, the equilibrium position $R \simeq R_{\mathrm{T}}$ can be lost, meaning the bubble can be pushed towards the outer region of the star, see Sec. 4.2. If such an instability takes place, the evolution of the bubble is no longer quasi-static, but rather the minimization of the energy of the system becomes a time-dependent problem that can be simply described by a time-dependent bubble radius, $R \to R(t)$, which quickly approaches relativistic speeds. Depending on how fast the potential barrier reappears with radius, the instability cannot be stopped and the bubble expands beyond the star. Specifically, we find that the bubble *escapes* if

$$\frac{\Delta\sigma}{\Delta R_{\mathrm{T}}} \lesssim \epsilon \,, \tag{19}$$

where $\Delta\sigma$ is the difference between the tension of bubble wall at $R \simeq R_{\mathrm{T}}$ and $R \gtrsim R_{\mathrm{S}}$. The fact that the wall tension changes as it propagates through the star is one of the unique aspects of the bubble dynamics at finite density. In particular, it gives rise to an extra force that prevents the bubble from escaping the star unless $\epsilon$ is large enough. While for a bubble connecting to a shallow metastable minimum the condition Eq. (19) is readily satisfied (given Eq. (18) is), it is harder in the case of a deep minimum, because of the significant increase of the wall tension, being eventually dominated by the large barriers of the potential in vacuum. The discussion above is visualized in Fig. 2.

## 4.1 Formation: linear potential approximation

Let us start by considering the classical formation of a bubble in a star where the critical density is reached. In order not to unnecessarily complicate the discussion, let us assume that the in-density potential can be well approximated by the rolling term only, i.e. that due to the suppression of $\Lambda_{\mathrm{B}}^4(n) = \Lambda_{\mathrm{B}}^4(1 - \zeta(n))$ we can neglect the barrier term,

$$V(\phi, n > n_c) \simeq -\mu^2 f \phi \,, \tag{20}$$

where recall that in Eq. (14) we have identified $\mu^2 \sim \Lambda_{\mathrm{R}}^4/f^2$ as the scale that characterizes the scalar profile. An exact solution to the scalar EOM with a linear potential is

$$\phi(r) = \frac{\mu^2 f}{6}(R_{\mathrm{T}}^2 - r^2) + \phi_- \,, \quad r \leqslant R_{\mathrm{T}}, \qquad \text{(proto-bubble)} \tag{21}$$

with boundary conditions $\phi'(0) = 0$ and $\phi(R_{\mathrm{T}}) = \phi_-$. We then simply take $\phi(r \geqslant R_{\mathrm{T}}) = \phi_-$. We find that the proto-bubble is of size $R = R_{\mathrm{T}}$ and the field displacement at its center, $\Delta\phi(0) \equiv \phi(0) - \phi_-$, is given by

$$\frac{\Delta\phi(0)}{f} = \frac{(\mu R_{\mathrm{T}})^2}{6} \,. \tag{22}$$

This situation is explicitly depicted in the second panel of Fig. 2. Eq. (21) constitutes a good a priori description of the scalar profile as long as the system is small enough that the in-density minimum, $(\phi_+)_n$, is not reached, i.e.

$$\frac{\Delta\phi(0)}{(\phi_+)_n - \phi_-} \lesssim 1 \,. \tag{23}$$

We recall that in general $(\phi_+)_n > \phi_+$, see the discussion around Eq. (7).

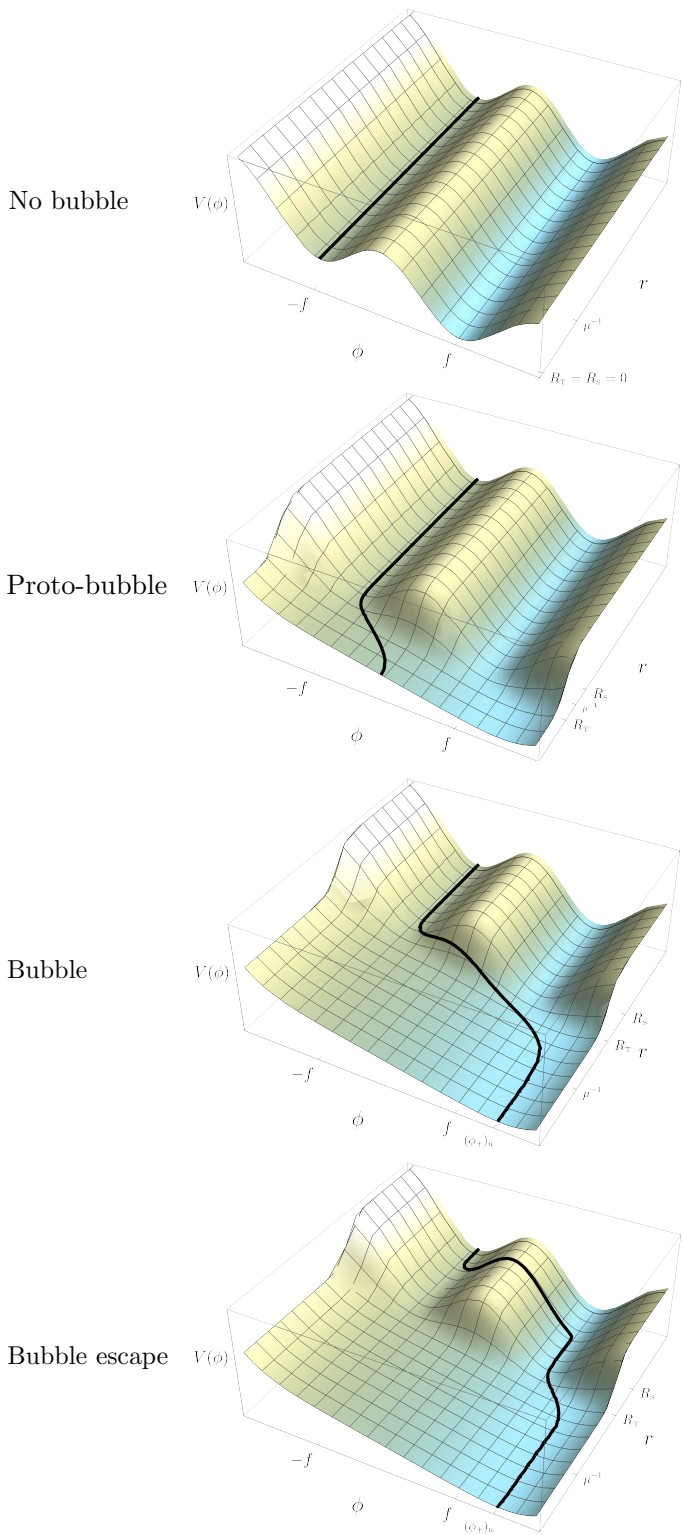

Figure 2: Quasi-static evolution of the in-density potential and scalar field profile, from no star to, as the star grows, the formation of the proto-bubble, complete formation of the bubble, and eventual bubble escape.

It is important to point out here that the quasi-static description of the proto-bubble can break down as soon as $\phi(0) \sim \phi_+$, as we discuss in Sec. 4.2. In this regard, Eq. (22) implies that any system, independently of its density profile or maximum density at its core, must have a minimum size in order for $\phi(0) \gtrsim \phi_+$, given by

$$R_{\text{T}} \gtrsim \mu^{-1}, \tag{24}$$

where we have neglected $O(1)$ factors.

The solution Eq. (21) can be extended to the situation in which the in-density minimum is reached somewhere inside the star, at $r = R_i < R_{\text{T}}$. In that region the potential exhibits a minimum, and consequently the scalar field remains pinned at $(\phi_+)_n$. This is depicted in the third panel of Fig. 2, where we have chosen a core density such that $(\phi_+)_n$ is only slightly larger than $\phi_+$. The scalar profile is well approximated by

$$\phi(r) = \begin{cases} (\phi_+)_n & r < R_i \\ -\frac{\mu^2 f}{6}(r - R_i)^2 + (\phi_+)_n & R_i < r < R_{\text{T}} , \qquad \text{(bubble)} \\ \phi_- & r > R_{\text{T}} \end{cases} \tag{25}$$

where in the intermediate region, $r \in [R_i, R_{\text{T}}]$, we have used the solution of the EOM with the linear potential Eq. (20), shifted it by $r \to r - R_i$, and required $\phi'(R_i) = 0$, $\phi(R_i) = (\phi_+)_n$; further matching to $\phi(R_{\text{T}}) = \phi_-$ fixes the value of $R_i$, or equivalently the width of the bubble wall

$$x \equiv \frac{R_{\text{T}} - R_i}{R_{\text{T}}} \simeq \frac{\sqrt{6}}{\mu R_{\text{T}}} \sqrt{\frac{(\phi_+)_n - \phi_-}{f}} . \tag{26}$$

Of course, in order for $R_i > 0$, $R_{\text{T}}$ needs to be large enough as to allow the field to reach the minimum at finite density. In other words, the requirement that $x < 1$ implies

$$R_{\text{T}} \gtrsim \frac{\sqrt{6}}{\mu} \sqrt{\frac{(\phi_+)_n - \phi_-}{f}} . \tag{27}$$

A scalar field profile for which this condition is satisfied is shown in Fig. 3, for a choice of central density not much larger than the critical density.

For an increasingly larger system, yet with with a core density fixed such that $(\phi_+)_n$ remains constant, the bubble wall becomes thinner, i.e. $x \ll 1$ when $\mu R_{\text{T}} \gg 1$. In this thin-wall limit, the energy of the bubble, Eq. (17), can be approximated by a volume and a surface term [1],

$$E(R) \simeq -\frac{4\pi}{3} R^3 \epsilon + 4\pi R^2 \sigma , \tag{28}$$

where $\epsilon$ is the (potential) energy difference between the in-density and in-vacuo field values, while $\sigma$ is the bubble-wall tension. For our simple scalar profile these read

$$\epsilon = \mu^2 f((\phi_+)_n - \phi_-) \gtrsim \Lambda_{\text{R}}^4, \tag{29}$$

$$\sigma = \frac{4}{3}\sqrt{\frac{2}{3}}((\phi_+)_n - \phi_-)\sqrt{\epsilon} \gtrsim \Lambda_{\text{R}}^2 f . \tag{30}$$

In Eq. (28) we have traded $R_{\text{T}}$ with $R$, since we are assuming that during the formation of the bubble its wall sits at $R \simeq R_{\text{T}}$; in Sec. 4.2 we discuss under which circumstances such an equilibrium is lost, i.e. $R > R_{\text{T}}$. Also, we have implicitly assumed that $(\phi_+)_n$ is constant below $R_i$, i.e. that the density does not significantly change for $r < R_i$. Both the inequalities in Eqs. (29), (30) follow from $(\phi_+)_n > \phi_+$, after neglecting $O(1)$ factors. These correspond to the minimum values of the potential energy and tension of a fully formed bubble. As expected, we find $\epsilon \gtrsim |\Delta V(\phi_+, n)| = -\Delta\Lambda$, where recall that $\Delta\Lambda$ is the energy difference between the false

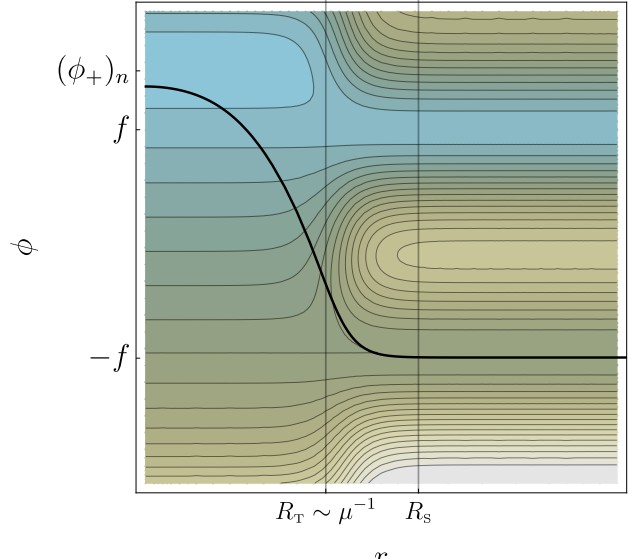

Figure 3: Scalar profile for $\mu R_{\mathrm{T}} \gtrsim 1$ on top of contours of the scalar potential.

and true ground states, Eq. (8). In addition, let us point out that the condition Eq. (27) can be understood from energy considerations, as the requirement that the (mean) field gradient is small enough, $\frac{1}{2}\langle\phi'^2\rangle \sim ((\phi_+)_n - \phi_-)^2/R_{\mathrm{T}}^2 \lesssim \epsilon$. In this regard, note also that the tension is dominated by the field displacement, $\sigma \sim ((\phi_+)_n - \phi_-)^2/(xR_{\mathrm{T}})$ [38].

In App. A we reproduce the above scalings with a simpler linear profile approximation, where we do not need to assume that the potential is well described by a linear slope only. In particular, we can keep the subdominant barrier term and we find that, while leaving $\epsilon$ unchanged, it gives a corrections to the tension of the bubble wall that scales as

$$\frac{\Delta\sigma}{\sigma} \sim \frac{\Lambda_{\mathrm{B}}^4(n)}{\Lambda_{\mathrm{R}}^4} \simeq \frac{1 - \zeta(n)}{1 - \zeta_c}. \tag{31}$$

This becomes negligible when $\zeta \to 1$, that is also when $(\phi_+)_n \gg \phi_+$, see Eq. (7). On the other hand, when the density is not much above critical, the correction is parametrically $O(1)$. Nevertheless, the most important effect of the potential barriers arises when we consider a bubble whose wall is beyond the transition radius, i.e. $R > R_{\mathrm{T}}$, as we discuss in the following.

## 4.2 Dynamics: escape vs equilibrium

In the previous discussion we worked under the assumption of a nearly-static bubble, which slowly grows with time only due to the increase in size of the star (or more accurately, due to the increase in size of the transition radius $R_{\mathrm{T}}$ where the critical density is reached). Here we show that in fact this adiabatic description can break down as soon as the star is (dense and) large enough that the field displacement inside it reaches the position of the true minimum in vacuum.

There are several ways to understand the origin of this instability. Qualitatively, for the potentials we are considering, finite density effects allow for the local minimum in vacuum to be continuously (i.e. classically) connected to the true minimum. This is because the in-vacuo potential barrier between them disappears in some region of the star ($r < R_{\mathrm{T}}$), see the right panel of Fig. 4. Once this region is large enough such that $\Delta\phi(0) \gtrsim \phi_+ - \phi_- \approx 2f$, it may become energetically favourable for the tail of the field profile, which extends outside the star,

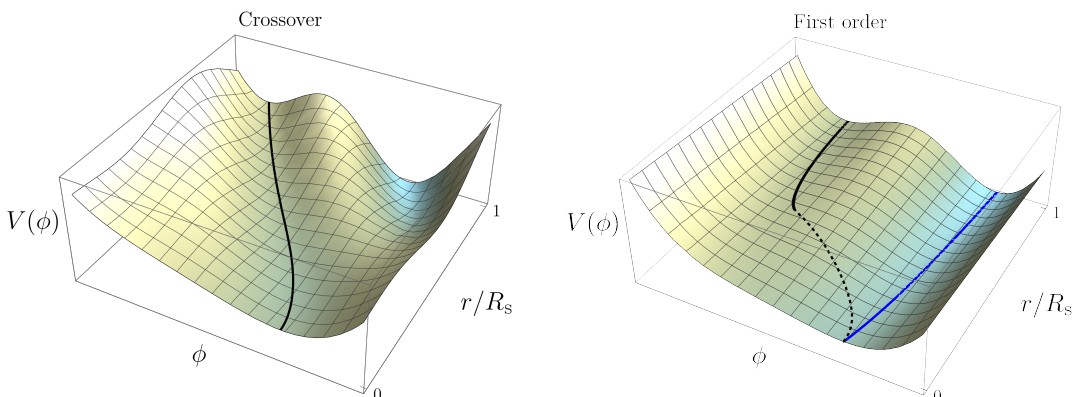

Figure 4: Crossover (left) versus first order phase transition (right) induced by a dense system (spherically symmetric and of finite size). For both cases the potential is shown as a function of radius, with $r/R_s = 0$ the center of the star. The black solid lines illustrate the scalar profile starting from a given in-vacuo ($r/R_s > 1$) minimum and following it inside the star. For a first-order phase transition, the black line stops where this minimum ceases to exist. The dashed line then illustrates the field profile that connects to the minimum within the star. The profile unavoidably passes through regions where $dV/d\phi \neq 0$, implying there are effective forces acting on the field. These forces give rise to the possibility that the initial scalar profile (black) classically changes to a new minimum in vacuum (blue).

to be pushed over the potential barrier. This effectively leads to a first-order phase transition in the form of a bubble escaping the star. This is in contrast with other types of potentials with metastable minima, such as that shown in the left panel of Fig. 4, where even at finite density there is always a potential barrier between the two minima. This class of potentials does not allow for a classical path connecting them, and therefore leads to a smooth cross-over to a different in-density minimum.[6]

Let us note that the discussion is focussed on field displacements that are at least of the order of the field separation between the local and true minimum in vacuum. This is because, at least qualitatively, a bubble with $(\phi_+)_n \sim \phi_+$ captures all the non-trivial dynamics of the phase transition. In the following we focus on such a case, which corresponds to maximal densities of the order of the critical density. A discussion of the bubble dynamics for $(\phi_+)_n \gg \phi_+$, is deferred to App. D.

In order to quantitatively understand the dynamics of induced first-order phase transitions, we resort to the description of the scalar bubble wall as a particle in $d = 1 + 1$ dimensions. While this is a standard treatment when studying the dynamics of bubbles in vacuum or at finite temperature (see e.g. [39]), here we adapt it to the finite density environment, crucially including a position-dependent tension, $\sigma(R)$. The Lagrangian for the time-dependent bubble-wall position $R(t)$ is given by

$$\mathcal{L} = -\mathcal{M}(R)/\gamma - \mathcal{V}(R),\tag{32}$$

where $\gamma = 1/\sqrt{1 - \dot{R}^2}$. In the thin-wall approximation, $x \ll 1$, where the particle description

---

[6]Even with non-vanishing barriers, finite density effects could lead to a significant increase in the tunneling probability to the true minimum, thus seeding a quantum first-order phase transition. We discuss this possibility in Sec. 4.4 since it is of relevance as well for our potential whenever densities remain below critical, i.e. $n(r) < n_c \forall r$.

best applies, we have

$$\mathcal{M}(R) = 4\pi \int_{R(1-x)}^{R} dr\, r^2 \left[ \frac{1}{2}\phi'^2 + \Delta V(\phi, n(r)) \right] \equiv 4\pi R^2 \sigma(R), \tag{33}$$

$$\mathcal{V}(R) = -\frac{4\pi}{3}R^3 \Delta\Lambda \equiv -\frac{4\pi}{3}R^3 \epsilon. \tag{34}$$

Several comment are in order regarding the bubble mass and potential at finite density. First, the bubble's energy given in Eq. (28) is precisely the Hamiltonian associated with Eq. (32) in the static limit $\dot{R} = 0$. Second, from the integral expression of $\mathcal{M}(R)$, it is clear that in the thin-wall limit the bubble wall is only sensitive to the density at $r = R$. Therefore, as the bubble moves through the star, its tension changes due to the changing density.[7] Since the bubble is born with $R \simeq R_{\scriptscriptstyle T}$, from Eq. (30) with $(\phi_+)_n \sim \phi_+$ we have

$$\sigma(R \simeq R_{\scriptscriptstyle T}) \sim \Lambda_{\scriptscriptstyle R}^2 f. \tag{35}$$

Recall that for the bubble to have been fully formed, $R_{\scriptscriptstyle T}$ needs to satisfy Eq. (27), which for $(\phi_+)_n \sim \phi_+$ reads $R_{\scriptscriptstyle T} \gtrsim \mu^{-1}$. Finally, $\mathcal{V}(R)$ is controlled by the potential energy difference between the two sides of the bubble wall, which from Eq. (29) with $(\phi_+)_n \sim \phi_+$ is given by

$$\epsilon \sim \Lambda_{\scriptscriptstyle R}^4. \tag{36}$$

The equation motion of the bubble wall reads

$$\sigma \ddot{R}\gamma^3 = \epsilon - \gamma\left(\frac{2\sigma}{R} + \sigma'\right), \qquad \sigma' = \frac{d\sigma}{dR}. \tag{37}$$

Since we are mainly interest in the dynamics of the bubble right after its formation, we concentrate on the non-relativistic limit, i.e. we set $\gamma = 1$. The right hand side of Eq. (37) is the sum of forces (pressures) acting on the bubble wall. The potential energy difference between the two sides of the wall pushes it outwards. The second and third terms are associated with the tension of the wall, both pushing it inwards. In particular, the change in tension $\sigma'$ is positive, since densities decrease with $R$ and in turn the potential barriers, controlled by $\Lambda_{\scriptscriptstyle B}^4(n)$, reappear and increase towards its vacuum value outside the star.

In order to understand the behaviour of $\sigma(R)$, let us first recall that when the bubble is just formed, the tension is dominated by the field displacement, see Eq. (30). This implies that only the contribution to the tension from the barrier, estimated in Eq. (31), leads to an increasing tension with $R$. For bubbles connecting shallow minima, $\delta^2 \ll 1$, this increase is small between $R_{\scriptscriptstyle T}$ and $R_{\scriptscriptstyle S}$,

$$\sigma(R_{\scriptscriptstyle S}) - \sigma(R_{\scriptscriptstyle T}) \sim f\Lambda_{\scriptscriptstyle R}^2\delta^2. \qquad \text{(shallow)} \tag{38}$$

In contrast, for deep minima, $\delta^2 \approx 1$, the tension goes from being displacement-dominated at $R \simeq R_{\scriptscriptstyle T}$, to barrier-dominated towards the end as well as outside of the star $R \simeq R_{\scriptscriptstyle S}$. There we can use the standard thin-wall approximation to compute the tension [1],

$$\sigma(r \simeq R_{\scriptscriptstyle S}) \simeq \int_{-f}^{f} d\phi \sqrt{2V(\phi)} \simeq \frac{2}{3}\Lambda_{\scriptscriptstyle B}^2 f, \qquad \text{(deep)} \tag{39}$$

---

[7]We are implicitly assuming that the width of the wall is the smallest scale in the system. If this were not the case, we would expect finite-size effects in the form of e.g. deformations of the bubble. However, these would lead to at most $O(1)$ corrections to our already approximate analytical results, leaving our qualitative conclusions unchanged.

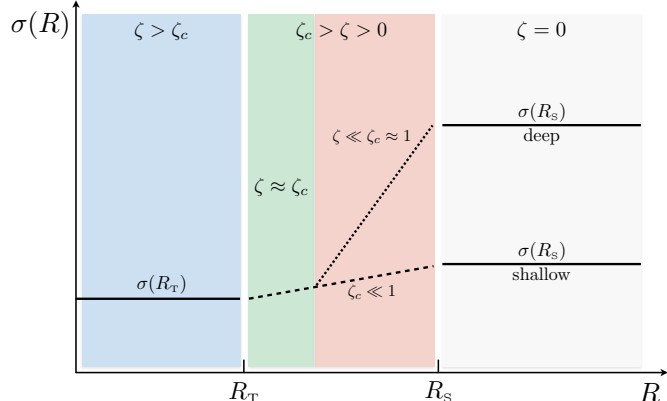

Figure 5: Sketch of the relevant regions of star for what concerns the bubble-wall tension. Dashed and dotted lines do not necessarily represent the functional form of $\sigma(R)$. In the green region the tension is dominated by the field displacement, while in the red region the barriers come to dominate. Note that for $\zeta_c = \delta^2 \ll 1$ (i.e. shallow minimum), there is in fact no red region.

and $\sigma(R_S) - \sigma(R_T) \simeq \sigma(R_S)$. In addition, let us note that the bubble gets thinner when the barrier term dominates the tension. The bubble wall tension, as a function of its location, is schematically summarized in Fig. 5 for both the shallow and deep minimum cases.

Before moving to the detailed discussion of how the changing tension affects the dynamics of the bubble wall, let us note that in Eq. (37) we have ignored the effect of the gravitational force of the star on the bubble wall. In App. B we discuss such a force, showing that while for neutron stars it could be quantitatively relevant at some stage during the expansion, it does not qualitatively change the picture presented here.

Having established the behaviour of the tension from $R_T$ to $R_S$, let us understand the dynamics of the bubble wall. Right after the formation of a thin-wall bubble at $R \simeq R_T \gtrsim \mu^{-1}$, the particle description Eq. (37) applies. One then automatically finds $\ddot{R} > 0$ right before the transition region, since we can simply assume that $\sigma'$ vanishes for $R < R_T$, that is $\sigma'(R_T^-) = 0$. The acceleration would remain positive in the limit that the force due to the change in tension vanished for any $R$, $\sigma' \to 0$; in this limit the bubble would expand indefinitely, in particular beyond the star. In the opposite limit, in which $\sigma'$ is very large just past the edge of the transition region, that is $\sigma'(R_T^+) \to \infty$, the wall could not expand and therefore it would remain at an equilibrium radius $R = R_{eq} = R_T$ (and the bubble would only grows if $R_T$ kept increasing). Clearly, a realistic situation lies in between these two limits, and it depends on how fast the density profile and thus the tension changes from $R_T$ to the end of the star. This discussion gives us a qualitative understanding of why the bubble might generically be found in an equilibrium position at $R_S > R > R_T$. In a similar fashion, we can understand under which conditions the bubble escapes from the star. In the limit that the star has grown so large that the transition region starts at a radius much larger than the one needed to form the bubble, i.e. $R_T \gg \mu^{-1}$, we have $\epsilon \gg 2\sigma(R_T)/R_T$. Then, it follows from the equation of motion that the bubble wall would continue to accelerate for $R > R_T$ as long as $\epsilon > \sigma'$. In the opposite limit, in which $R_T \simeq \mu^{-1}$, we have $\epsilon \to 2\sigma(R_T)/R_T$ and the additional force due to $\sigma'$ would be enough to forbid its expansion. These different limits lead us to the conclusion that for a sufficiently large star, satisfying $R_T \gtrsim \sigma(R_T)/[\epsilon - \sigma'(R_T)]$, the system is unstable and the bubble escapes if

$$\epsilon \gtrsim \kappa \, \sigma'_{max}, \qquad (40)$$

where $\sigma'_{max}$ is the maximum value of $\sigma'$ and $\kappa = O(1)$. This condition is explicitly verified by

our numerical simulations as well as in App. C, where we investigate Eq. (37) in the simplest case of a constant $\sigma'$, finding $\kappa = 3$. Once again, (a version of) this condition can be expected to hold in general, on the basis that the standard force due to the surface tension becomes irrelevant at large $R$, leaving the variation of the tension as the only relevant force to determine if the bubble does or does not escape from the star.

### 4.3 Summary: formation and escape conditions

Given that the change in the wall tension is very different for a bubble connecting shallow or deep minima in vacuum, let us explicitly summarize for each case the conditions under which the bubble forms and escapes from the star.

For a shallow bubble, $\delta^2 \ll 1$, we find as formation and escape conditions, respectively

$$R_{\mathrm{T}} \gtrsim \frac{f}{\Lambda_{\mathrm{R}}^2} \quad \text{and} \quad \Delta R_{\mathrm{T}} \gtrsim \frac{f}{\Lambda_{\mathrm{R}}^2} \delta^2, \qquad \text{(shallow)} \tag{41}$$

up to irrelevant $O(1)$ factors. Note that since $\sigma'$ is suppressed by $\delta^2$, as shown in Eq. (38), the escape condition is easier to satisfy than the condition for formation. This is unless, contrary to the expectation from generic density profiles, $\Delta R_{\mathrm{T}}$ is anomalously small. In terms of the mass of the scalar in vacuum, Eq. (3), these two conditions read as $m_\phi R_{\mathrm{T}} \gtrsim \sqrt{\delta}$ and $m_\phi \Delta R_{\mathrm{T}} \gtrsim \delta^{5/2}$.

For a bubble connecting deep minima, $\delta^2 \approx 1$, the rate of change of the tension is determined by the tension in vacuum, $\sigma' \sim \sigma(R_{\mathrm{S}})/\Delta R_{\mathrm{T}}$, as shown in Eq. (39). Therefore, we find the following conditions for the formation and escape of a deep bubble, respectively

$$R_{\mathrm{T}} \gtrsim \frac{f}{\Lambda_{\mathrm{R}}^2} \quad \text{and} \quad \Delta R_{\mathrm{T}} \gtrsim \frac{f}{\Lambda_{\mathrm{R}}^2} \frac{1}{\sqrt{1-\delta^2}}, \qquad \text{(deep)} \tag{42}$$

up to $O(1)$ factors. As expected, it is generically much more difficult for a bubble connecting deep minima to transverse the transition region and expand beyond the star. Besides, while the condition for formation is formally the same as for shallow minima, let us recall that $\zeta_c = \delta^2 \approx 1$ generically implies that much larger densities are needed in this case. In terms of the mass of the scalar in vacuum, Eq. (3), the two conditions in Eq. (42) read as $m_\phi R_{\mathrm{T}} \gtrsim 1/\sqrt{1-\delta^2}$ and $m_\phi \Delta R_{\mathrm{T}} \gtrsim 1/(1-\delta^2)$.

### 4.4 Classical vs quantum

To conclude this section, we wish to investigate the possibility that, even when the system is not dense enough as to allow for a classical transition between the local and true minimum, finite density could still lead to a much shorter quantum-mechanical lifetime of the metastable minimum compared to the one in vacuum. This is reminiscent of the idea that black holes or compact objects can act as seeds for false vacuum decay, due to their strong gravitational fields, see e.g. [40–46].

Indeed, up until this point we did not care about the lifetime of the false vacuum, implicitly assuming that it was sufficiently large. The decay rate per unit volume is determined by the bounce action, $\Gamma/\mathcal{V} = A e^{-S_{\mathrm{B}}}$ [1, 2]. By definition, in the case where the metastable minimum is deep, the thin-wall approximation holds. The action is well approximated by $S_{\mathrm{B}} \simeq (27/2)\pi^2 \sigma^4/\epsilon^3$, which given the in-vacuo tension Eq. (39) and $\epsilon = -\Delta\Lambda \simeq \frac{2}{3\sqrt{3}}\Lambda_{\mathrm{R}}^4$, results in

$$S_{\mathrm{B}} \simeq 27\sqrt{3}\pi^2 \left(\frac{f}{\Lambda_{\mathrm{B}}}\right)^4 \frac{1}{(1-\delta^2)^3}. \qquad \text{(deep)} \tag{43}$$

Since $\delta^2 \approx 1$ for a deep minimum, the bounce action is generically large and the decay rate extremely suppressed. For a shallow minimum, we can estimate the action by considering

$\sigma \sim \Delta\phi^2/\Delta R$ with $\Delta R \sim \Delta\phi/\sqrt{\epsilon}$, which leads to $S_{\text{B}} \sim \pi^2 \Delta\phi^4/\epsilon$. We therefore find,[8]

$$S_{\text{B}} \sim 24\pi^2 \left(\frac{f}{\Lambda_{\text{B}}}\right)^4. \qquad \text{(shallow)} \tag{44}$$

While for the same value of the ratio $f/\Lambda_{\text{B}}$ the bounce action is smaller in the shallow than in the deep case, this is not the comparison we really care about. Instead, let us assume that the local minimum is, for all practical purposes, stable in vacuum. This fact can dramatically change in a dense system only in the case of a deep minimum (even before a classical transition is allowed). This is clear since for a shallow minimum $S_{\text{B}}(n < n_c) \simeq S_{\text{B}}(0)$, while for a deep one

$$\frac{S_{\text{B}}(n < n_c)}{S_{\text{B}}(0)} \simeq [1 - \zeta(n)]^2, \qquad \text{(deep)} \tag{45}$$

which is much smaller than one if $\zeta \approx 1$ (yet $\zeta < \zeta_c = \delta^2$). Generically, the decay rate can only be sufficiently fast compared to the lifetime of star if the bounce action is not very large, which drives us towards the non-perturbative regime for the scalar quartic coupling $\lambda \equiv (\Lambda_{\text{B}}/f)^4$.[9] Nevertheless, this conclusion could well be specific to the type of false vacua we are taking as case study, thus one could imagine other scalar potentials where, being sensitive to finite density (either of SM degrees of freedom or beyond, e.g. dark matter), their local minima have much smaller lifetimes in a dense system. Additionally, let us note that the corresponding seeded nucleation of bubbles of the true ground state would generically not take place during the formation of the star. On the contrary, one would expect $1/\Gamma \gg T_{\text{s}}$, while still being shorter than the the typical lifetime of the star, $1/\Gamma \ll \mathcal{T}$, such that the decay probability $\mathcal{T}\Gamma = O(1)$, see footnote 4.4. This raises the possibility of a latent phase transition that could take place at any time.

Finally, let us point out that in the computation of the bounce action at finite density, we have assumed the system is large and homogeneous enough as for the effects of a non-trivial density profile or a spatial boundary to be negligible. We can phrase this as the requirement that $R_0 \ll R_{\text{s}}$, where $R_0 = 3\sigma/\epsilon$ is the radius of the nucleated bubble. For a deep minimum, this translates into $m_\phi R_{\text{s}} \gg 1/(1-\delta^2)$, which coincides with the condition for the escape of a deep, classically formed, bubble, see Eq. (42). It would be interesting to further study, beyond these simple approximations, the process of quantum bubble nucleation in finite-size dense systems [48, 49].

---

[8]More refined estimates can be easily derived for potentials where the barrier is negligible, see e.g. [47]. Nevertheless, our conclusions will not depend on such a refinement.

[9]Focussing on neutron stars as nucleation seeds, we can roughly estimate the requirement for an $O(1)$ tunnelling probability by taking into account the volume of all neutron stars in the observable universe since the time of star formation until today (since such stars are stable in isolation). The volume of all neutron stars in the observable universe is roughly $\mathcal{V}_{\text{NS}} \sim R_{\text{NS}}^3 N_{\text{NS/G}} N_{\text{G}} \sim 10^{32}\,\text{m}^3$, where $R_{\text{NS}} \sim 10\,\text{km}$ is the typical radius of a neutron star, $N_{\text{NS/G}} \sim 10^9$ the number of neutron stars per galaxy (like ours) and $N_{\text{G}} = 10^{11}$ the number of galaxies. Since star formation happened relatively early, the relevant time scale is $\mathcal{T} \sim 1/H_0 \sim 10^{17}\,\text{s}$. Requiring that $\mathcal{T}\Gamma \sim \mathcal{O}(1)$ and using Eq. (43), we find

$$\frac{\lambda(1-\delta^2)^3}{(1-\zeta)^2} \sim \frac{27\sqrt{3}\pi^2}{\log(f^4 \mathcal{T} \mathcal{V}_{\text{NS}})} \gtrsim 1, \tag{46}$$

where the last inequality arises from assuming $f < M_{\text{Pl}}$. Note that we also used the NDA estimation for the exponent coefficient $A \sim f^4$. In the best case scenario the metastable minimum is not very deep, e.g. $\delta \approx 0.8$. Demanding a mild reduction of the potential barrier in density, $\zeta \approx 0.5 < \delta^2$, in order to catalyse the decay we need $\sqrt{\lambda} \gtrsim 2.3$, which is certainly not a weak coupling.

# 5 Phenomenological implications

In this section we discuss the phenomenological consequences of the expansion, beyond the dense object, of a bubble of the true vacuum. The main model-independent signature of such a seeded phase transition is a change of the vacuum energy of the universe, $\Lambda$, or equivalently a change of the cosmological dark energy density, $\rho_\Lambda$ (with equation of state parameter $\omega = -1$).[10]

A particularly interesting trademark of these phase transitions is that they take place relatively late in the history of the universe. As explained in the previous section, the bubble forms, expands and eventually escapes along with the formation of the star. Therefore, if a phase transition of this sort can happen, it took place at the onset of star formation. The first stars were born around the epoch of galaxy formation, thus at redshifts $z = z_s \sim 10$ [50]. This then implies that the universe underwent a change of $\rho_\Lambda$ between recombination, $z \sim 10^3$, and the late universe, $z \lesssim 1$. Note that we are assuming that at redshifts $z \sim 1$ (associated with late-time cosmological measurements) the universe already transitioned successfully to the true ground state. The change in the dark energy content of the universe can thus best be probed by comparing CMB measurements versus local measurements (SNe, baryon acoustic oscillations or large-scale structure) of the expansion rate of the universe. Such a comparison depends on the fate of the bubbles, for instance if the phase transition proceeds via a single bubble or instead many bubbles are formed all over the universe (from as many stars) that subsequently collide and transfer at least an $O(1)$ fraction of the kinetic energy of their walls into radiation. Providing a precise answer to this question is beyond the scope of this work. Instead, below we work out simple cosmological constraints on how much the energy budget of the universe can vary due to a late ($z \sim 10$) phase transition, to confirm our intuition that a change in the vacuum energy much larger than the current one is experimentally ruled out.

A too large change in vacuum energy leads to constraints on the parameters of the scalar potential. To make this point clear, let us note that the change in vacuum energy is given by $\Delta\Lambda = -\epsilon \sim -\Lambda_R^4$, and the rolling scale enters both the conditions for formation and escape of a bubble of the true vacuum, see Eqs. (41), (42). Then, assuming the existence of stars with densities above critical, $n > n_c$, the condition for formation of a bubble with $R_T \sim R_s$, as expected for most stellar profiles, implies

$$-\Delta\Lambda \gtrsim \left(\frac{f}{R_s}\right)^2 \approx \Lambda_0 \times 10^{15} \left(\frac{f}{10\,\text{TeV}}\right)^2 \left(\frac{10\,\text{km}}{R_s}\right)^2 , \tag{47}$$

where $\Lambda_0 \approx (2.3\,\text{meV})^4$ is the value of the vacuum energy inferred from $\Lambda$CDM, and we have fixed $R_s$ to the typical radius of a neutron star as an example. If such type of bubbles could have escaped from neutron stars, the corresponding change in the vacuum energy would be in gross contradiction with experimental data. Note that a similar region of parameter space is realized in e.g. relaxion models [10].

However, for much smaller values of $f$, or if we were to consider much larger astrophysical bodies (the largest stars known have $R_s \sim 10^3 R_\odot$), astronomical structures, or even dense objects beyond the SM (such as dark stars), the change in the dark energy density could be much smaller. In particular, the very first stars to form ($z \sim 20-30$), the so-called Pop. III stars, are believed to have been supermassive $M \gtrsim 100\,M_\odot$ and as large as $R \sim 10^3 R_\odot$, and therefore could be interesting candidates [51]. Likewise, the largest stars observed are red giants with radii up to $R \sim 10^3 R_\odot$ and masses $M \sim 10\,M_\odot$. The corresponding nucleation of bubbles of the true vacuum and subsequent phase transition could then be an experimentally viable and very interesting phenomenon, which could be detected in the near future given the expected

---

[10]In the following we exclude the possibility of an adjustment mechanism for the cosmological constant. Such a mechanism could interfere with the formation or escape of the bubble.

increase in precision of many current and planned cosmological observatories. Whether or not this type of phase transition could lead to interesting gravitational wave signatures is an interesting question, which we leave for future investigation along with the study of the corresponding cosmological dynamics.

Amusingly, if the phase transition proceeds via quantum tunneling, as we have argued in Sec. 4.4, a recent creation of a true vacuum bubble could lead to other, more direct, experimental signatures: since the bubble interacts with SM matter, gravitationally at the very least, the effects of a (non-percolated) bubble wall passing through Earth could potentially be detected [52, 53].

Let us finally point out that seeded phase transitions with $\Delta\Lambda \lesssim \Lambda_0$ could impact our understanding of the landscape solution to the cosmological constant problem. Originally connected with the requirement for galaxies and stars to form [5], the cosmological constant was predicted to lie within a range a couple of orders of magnitude larger than the value actually observed as dark energy. In light of our late-time phase transitions, taking place precisely because structures form, this discrepancy could well be an accident associated with the sensitivity to finite density effects of a scalar potential with metastable minima (potentially many of them as in [54]).

## 5.1  Cosmological constraints

While it is beyond the scope of this work to examine in detail the cosmological and astrophysical constraints arising from a phase transition at the dawn of galaxy/star formation, let us briefly comment on simple arguments why a large change in the energy content of the universe is not experimentally viable.

From local measurements of the (accelerated) expansion of the universe, we know it is dark energy dominated, and in particular $\rho_r \ll \rho_\Lambda$ at $z \lesssim 1$, where $\rho_r$ is the energy density in radiation. If we assume that, at redshifts $z_s \sim 10$, an $O(1)$ fraction of the kinetic energy of the bubbles goes into radiation after they collide and percolate, then we find $\epsilon = \Delta\rho_r(z_s) \ll (1 + z_s)^4 \rho_{\Lambda_0} \approx 10^4 \rho_{\Lambda_0}$, which is inconsistent with e.g. Eq. (47).

Still, it would be preferable to proceed with minimal assumptions regarding the fate of the bubble. One relatively robust assumption is that today our Hubble patch is in the true vacuum, while it was not prior to star formation, that is $\rho_\Lambda(z > z_s) \neq \rho_{\Lambda_0}$. In this case, the most reliable test is to contrast late versus early universe measurements, something that has been actively pursued in recent years in light of the Hubble tension, the disparity between CMB and local determinations of the Hubble constant (see [55, 56] for recent discussions). Of particular relevance is the study in [57], where constraints on the size of an early dark energy content of the universe at the time of recombination are derived. The bounds are given as a function of the critical redshift $z_c$ where the dark energy starts to decay quickly, as $1/a^6$ (thus faster than radiation). Such a behaviour decreases the impact of this non-standard energy component at later times $z < z_c$, which we take as a good approximation towards independence from the fate of the bubble(s). Identifying $z_c = z_s$, the bound $\rho_\Lambda(z > z_s) \gtrsim 10^2 \rho_{\Lambda_0}$ is derived, two order of magnitude stronger than the crude bound we derived before. Although we expect that a proper analysis of the fate of the bubbles and its impact on cosmological observables would yield even stronger bounds, in this work we will take

$$-\Delta\Lambda \lesssim 10^2 \times \Lambda_0 \tag{48}$$

to set constraints on the parameters of the scalar potential Eq. (1).

For the bound Eq. (48) to apply, the conditions for a bubble of the true ground state to form and escape from the dense system must be satisfied. Let us recall that the first of these conditions is that densities need to be above the critical density, i.e. $n > n_c$, or more specifically

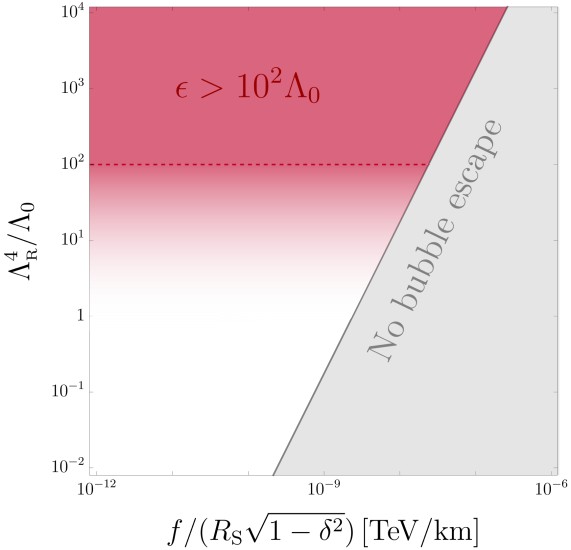

Figure 6: Region excluded by a density induced vacuum instability (shaded red) in the plane $(f/R_{\text{S}}\sqrt{1-\delta^2}, \Lambda_{\text{R}}^4/\Lambda_0)$, where $R_{\text{S}}$ is the typical radius of the (type of) star triggering the phase transition, i.e. where densities above critical are realized, $n > n_c$. The dashed line corresponds to the bound Eq. (48).

$$\zeta(n) > 1 - \frac{\Lambda_{\text{R}}^4}{\Lambda_{\text{B}}^4}, \tag{49}$$

see Eqs. (2), (6). Since in this work we do not focus on any specific scenario for the function $\zeta(n)$,[11] we simply assume that stars exist with $n > n_c$, and note that denser stars are typically smaller. The other conditions concern the formation and escape of the bubble, which are different for a shallow metastable minimum than for a deep one, see Eq. (41) and Eq. (42), respectively. These depend on either $R_{\text{T}}$ or $\Delta R_{\text{T}} = R_{\text{S}} - R_{\text{T}}$, which in turn depend on the density profile of the star. We will take $R_{\text{T}} \sim \Delta R_{\text{T}} \sim R_{\text{S}}$ as a generic expectation for stars where the core density is not very close to the critical one, as discussed in Sec. 3. Under this assumption, the strongest of the formation and escape conditions, for both shallow and deep minima, can be written as

$$\Lambda_{\text{R}}^4 \gtrsim \frac{f^2}{R_{\text{S}}^2} \frac{1}{1-\delta^2}. \tag{50}$$

We show the region of parameter space where this condition is satisfied in Fig. 6. Since a phase transition seeded by stars takes place in this region, the bound Eq. (48) applies, ruling out the corresponding part of it. Note that for a bubble connecting deep minima, Eq. (50) can be rewritten as $\Lambda_{\text{R}}^4 \gtrsim \Lambda_{\text{B}}^2 f/R_{\text{S}}$.

# 6 Conclusions

Could a phase transition have taken place in the universe due to the formation of stars? In this paper we explored this question by studying how false vacua change at finite density. Similar to the interactions with a thermal bath, the coupling of a scalar field to background matter can give rise to significant deformations of the scalar potential, to the point that a metastable

---

[11]Constraints on relaxion models, where $\zeta(n)$ can be explicitly computed, are presented in [19].

minimum present in vacuum disappears at finite density. This leads to the formation of a non-trivial scalar profile, a.k.a. a scalar bubble, where the maximum field displacement within is controlled by the size of the dense system relative to the characteristic scale of the in-density potential; if the star gets large enough, a classical path to a deeper minimum of the potential becomes accessible. Interestingly, we found that when this occurs, the bubble, originally confined within the star, can become unstable and expand beyond the star and extend to infinity! By means of simple analytic arguments, we have shown that the bubble cannot be contained within the star if the energy difference between the minima is large compared to how fast the potential barrier between them reappears towards the surface of the star. In other words, we have shown that if certain conditions regarding the properties of the metastable minimum and of the density profile are satisfied, stars can indeed act as seeds for a phase transition in the universe.

Our analysis of the fate of a false vacuum at finite density has been based on a tilted quartic potential, as in the classic work by Coleman [1]. This potential is characterized by the energy difference between the local and true minimum, the height of the potential barrier between them, and their separation in field space. Such a simple potential encodes the main features of local minima present in many scenarios beyond the SM. Specifically, our work can be extended to the relaxion [19], a mechanism to explain the smallness of the electroweak scale that relies on a closely-packed landscape of local minima, with barriers between that depend on the value of Higgs field thus sensitive to SM matter densities [10]. Other scenarios connected to the electroweak hierarchy problem or simply relying on the Higgs-portal, e.g. [11,12,23,58,59], should be investigated as well in light of our findings. In this regard, let us note that while we have focussed on scenarios where density affects the size of the potential barrier between minima, the analysis could be carried over to more general situations, e.g. by considering other scales to be density-dependent. Additionally, while we focused for concreteness on matter density, one should also consider other non-trivial backgrounds, such as an electro-magnetic field, as sources for the instability of the false vacuum [19].

Phase transitions triggered by dense systems such as stars must confront the experimental constraints that arise from the change in the energy of the vacuum at late cosmological times, $z \sim 10$, when star formation begins. Indeed, on the one hand the change in the ground state energy between the local and true vacuum is the key parameter that determines if a scalar bubble formed in a dense and large enough star is able to escape and propagate to infinity. On the other hand, early versus late cosmological measurements of the dark content of the universe constrain such a change. Nevertheless, we have shown that if the field distance between the minima is small enough or if the stars that can trigger the phase transition are very large, the phase transition could have taken place consistent with current cosmological data. Detailed cosmological and astrophysical constraints on these types of transitions, beyond the simple and likely too conservative bounds we have derived, deserves further investigation, in particular because of the relevance of scalar potentials with (many) false vacua for the electroweak hierarchy or the cosmological constant problems.

Finally, even though we focussed on classical transitions between minima, we have also shown how stars could act as a catalyzer where the tunneling probability of a false vacuum can be greatly enhanced. Although of a different, quantum-mechanical origin, once formed the dynamics of the corresponding scalar bubble would be described along similar lines as those presented here. The possibility of a seeded vacuum decay leaves us with another question: is it likely that a phase transition in the universe due to the formation of stars is soon to take place?

## Acknowledgments

The work of RB, JS, KS, SS and AW has been partially supported the Collaborative Research Center SFB1258, the Munich Institute for Astro- and Particle Physics (MIAPP), and by the Excellence Cluster ORIGINS, which is funded by the Deutsche Forschungsgemeinschaft (DFG, German Research Foundation) under Germany's Excellence Strategy – EXC-2094-390783311. RB is additionally supported by grants from NSF-BSF, ISF and the Azrieli foundation.

## A  Linear field profile approximation

The parametric dependence of the results in Sec. 4.1 can be reproduced by considering a simpler, linear approximation for the scalar profile (recall $\Delta\phi(0) \equiv \phi(0) - \phi_-$)

$$
\phi(r) = \begin{cases} \phi(0) & r < R_i \\ \phi(0) - \frac{\Delta\phi(0)}{R_{\text{T}} - R_i}(r - R_i) & R_i < r < R_{\text{T}} \\ \phi_- & r > R_{\text{T}} \end{cases} \quad \text{(bubble; linear)} \tag{51}
$$

and treating both $\phi(0)$ and $R_i$ as variational parameters determined by the minimization of the energy of the bubble $E(\phi(0), R_i)$, i.e. Eq. (17) with $R = R_{\text{T}}$. Expressing it in terms of $\Delta\phi(0)$ and the width $x = 1 - R_i/R_{\text{T}}$, the energy is given by

$$
E(\Delta\phi(0), x) = -E_0 \frac{\Delta\phi(0)}{f}\left[1 - \tfrac{3}{2}x + x^2 - \tfrac{1}{4}x^3 - \tfrac{3}{8}\frac{\alpha\Delta\phi(0)}{f x}\left(1 - x + \tfrac{1}{3}x^2\right)\right], \tag{52}
$$

where $E_0 = \frac{4\pi}{3}\mu^2 f^2 R_{\text{T}}^3$ and we have defined

$$
\alpha \equiv \frac{4}{(\mu R_{\text{T}})^2}. \tag{53}
$$

During the formation of the system, $R_{\text{T}}$ is small and therefore $\alpha \gg 1$. Minimization of the energy with respect to both $\Delta\phi(0)$ and $x$ yields $x = 1$ and

$$
\frac{\Delta\phi(0)}{f} = \frac{1}{\alpha}. \tag{54}
$$

Therefore, we find a proto-bubble ($R_i = 0$) in which the field displacement at the origin is $\Delta\phi(0)/f \sim (\mu R_{\text{T}})^2$, which is the result of an optimal balance between the gradient and potential energies. Parametrically, this matches the result in Eq. (22), albeit with a different numerical coefficient. As soon as the slowly-growing star is large enough that the in-density minimum $(\phi_+)_n$ can be reached, which happens when $\alpha \lesssim f/((\phi_+)_n - \phi_-)$, it should be energetically favorable for the profile to develop a core where the scalar value is fixed to $\phi(0) = (\phi_+)_n$. Then, minimization of the energy with respect to $x$ leads to

$$
x = \frac{1}{2}\sqrt{\frac{\alpha((\phi_+)_n - \phi_-)}{f}} + O(\alpha), \tag{55}
$$

This matches the result in Eq. (26), except for a numerical factor. Likewise, the energy of the bubble in the thin-wall limit $\alpha \ll f/((\phi_+)_n - \phi_-)$ is given by Eq. (28) where $\epsilon$ and $\sigma$ scale as in Eqs. (29), (30) respectively, $\epsilon = \mu^2 f((\phi_+)_n - \phi_-)$ and $\sigma = ((\phi_+)_n - \phi_-)\sqrt{\epsilon}$.

The linear profile Eq. (51) has the advantage that it is simple to estimate the importance of departures from the approximation of a linear potential, Eq. (1), we have worked under in

the main text. In particular, we can compute the effects of including the barrier term in Eq. (1) at finite density, i.e. with $\Lambda_{\text{B}} \to \Lambda_{\text{B}}(n)$. While $\epsilon$ remains unchanged in the thin-wall limit, the tension receives a correction

$$\frac{\Delta\sigma}{\sigma} = \frac{\sqrt{3}}{10} \frac{\Lambda_{\text{B}}^4(n)}{\Lambda_{\text{R}}^4}, \tag{56}$$

where we have assumed that the bubble is thin enough as to probe a fixed density.

## B  Gravitational force

In the equation of motion of the bubble, Eq. (37), we have neglected the gravitational force that the star exerts on the wall. While this does not change the conclusions we derived in the main text, it can lead to $O(1)$ numerical changes of the bubble's escape condition, at least for the densest stars, i.e. neutron stars.

In the non-relativistic and weak-field limits, the gravitational force of the star on the bubble wall per unit area (i.e. the pressure), is given by

$$F_{\text{G}}(R) = -\frac{1}{8\pi M_{\text{P}}^2} \frac{m(R)\sigma}{R^2}, \tag{57}$$

where $m(R)$ is the enclosed mass of the star and $\sigma$ the wall tension. Using a simple estimate for the neutron star number density $n \sim m_n^3$ and radius $R_{\text{NS}} \sim \sqrt{8\pi} M_{\text{P}}/m_n^2$, obtained by equating (Fermi-degeneracy) kinetic and gravitational energy densities and where $m_n$ is the neutron mass, we find $m(R) \sim 8\pi M_{\text{P}}^2 R^3/R_{\text{NS}}^2$. Therefore, for a neutron star

$$\text{NS}: \quad F_{\text{G}}(R) \sim \frac{\sigma R}{R_{\text{NS}}^2}, \tag{58}$$

while for less dense stars the gravitational force is much smaller, i.e. for white dwarfs it is suppressed by $m_e/m_p$. This additional force leads to a modification of the bubble wall equation of motion, in the non-relativistic limit (weak-field) and for $R \lesssim R_{\text{NS}}$

$$\sigma\ddot{R} \simeq \epsilon - \frac{2\sigma}{R}\left(1 + \frac{R^2}{2R_{\text{NS}}^2}\right) - \sigma', \tag{59}$$

which is subleading to the tension force except for $R \sim R_{\text{NS}}$. Likewise, once if the bubble leaves the star, the enclosed mass is the total mass of star and therefore for $R \gtrsim R_{\text{NS}}$

$$\sigma\ddot{R} \simeq \epsilon - \frac{2\sigma}{R}\left(1 + \frac{R_{\text{NS}}}{2R}\right), \tag{60}$$

which once again introduces an $O(1)$ change only when $R \sim R_{\text{NS}}$.

## C  Linear tension approximation

The simplest modelling of $\sigma(R)$, that is a constant $\sigma'$, allows us to analytically derive the condition Eq. (40). Let us then consider a linear increase of the tension with $R$, starting at $R_{\text{T}}$ and ending at $R_{\text{S}} = R_{\text{T}} + \Delta R_{\text{T}}$, thus with $\sigma' = [\sigma(R_{\text{S}}) - \sigma(R_{\text{T}})]/\Delta R_{\text{T}}$ constant. The equilibrium position of the bubble wall is determined by $\ddot{R}(R = R_{\text{eq}}) = 0$, and reads

$$R_{\text{eq}} = \frac{2[\sigma'R_{\text{T}} - \sigma(R_{\text{T}})]}{3\sigma' - \epsilon}, \qquad R_{\text{T}} > \sigma(R_{\text{T}})/\sigma' \quad \text{and} \quad 3\sigma' > \epsilon, \tag{61}$$

where the inequalities ensure that this is indeed an equilibrium position, i.e. with $E''(R_{\rm eq}) > 0$, where $E(R)$ is the energy of the bubble (note that $\ddot{R} \propto -E'$). For consistency, we should also require $R_{\rm eq} \gtrsim R_{\rm T}$, since that means that the bubble can in fact enter the transition region, where $\sigma' \neq 0$. This happens only if the star has grown large enough

$$R_{\rm T} > \frac{2\sigma(R_{\rm T})}{\epsilon - \sigma'}. \qquad \text{(entry transition region)} \qquad (62)$$

This condition is equivalent to the requirement $\ddot{R}(R_{\rm T}) \not< 0$,[12] and it only makes sense for $\epsilon > \sigma'$. If the condition Eq. (62) is not satisfied, it just means that $R_{\rm eq} = R_{\rm T}$ and the bubble is trapped inside the star. In addition, note that whenever the bubble is able to enter the transition region but the conditions in Eq. (61) are not satisfied, then the bubble automatically escapes the star, since there is no stable radius $R > R_{\rm T}$ for which $\ddot{R} = 0$ and $E'' > 0$. If instead the conditions in Eqs. (61), (62) are satisfied, then there is indeed an equilibrium position at $R_{\rm eq} > R_{\rm T}$, which increases as the star gets larger. This last fact generically leads to a smaller force from the term $2\sigma/R$ in Eq. (37). Eventually, the equilibrium condition is lost when the position of the wall reaches the outer edge of the star, i.e. $R_{\rm eq} \gtrsim R_{\rm S}$. This takes place when

$$R_{\rm T} > \frac{3\sigma(R_{\rm S}) - \sigma(R_{\rm T}) - \epsilon \Delta R_{\rm T}}{\epsilon - \sigma'}. \qquad \text{(exit transition region)} \qquad (63)$$

With the linear approximation for $\sigma(R)$ we then conclude that, as long as the volume energy of the bubble is larger than the rate of change of the tension, there is a minimum transition radius such that the bubble can permeate through the transition region, Eq. (62), and another for which the bubble can reach the surface of the star, Eq. (63). From that point outwards the bubble expands throughout the whole universe, since $\ddot{R}(R > R_{\rm S}) > 0$. Moreover, we also learn that if $\epsilon > 3\sigma'$, the only equilibrium position is $R_{\rm eq} = R_{\rm T}$, and this is lost as soon as the star is large enough as to satisfy Eq. (62). Importantly, let us note that when $\epsilon > 3\sigma'$, Eq. (62) is in fact approximately the same as the condition for the formation of the bubble, Eq. (54), thus in this case the formation and escape of the bubble take place simultaneously.

## D   Ultra-high densities

In Sec. 4.2 we centered our discussion of the bubble dynamics on the case where densities in the core of the star, while above critical, are not much larger than $n_c$. This is because a fully formed bubble for which the field at its center is $(\phi_+)_n \sim \phi_+$ already allows for the possibility of a classical phase transition to the true vacuum.

In this appendix we extend our analysis to the case of ultra-high densities, by which we mean $\zeta \to 1$. In this situation, the only minimum of the in-medium potential is found at $(\phi_+)_n \gg \phi_+$, see Eq. (7). As we explain in the following, we find that the escape of a bubble of the true vacuum can take place regardless of the scalar inside the star reaching the in-density minimum of the potential, i.e. $\phi(0) < (\phi_+)_n$, but it is enough that the field displacement is at least $\Delta\phi(0) \gtrsim \phi_+ - \phi_-$. As a matter of fact, if the star is large enough as to allow $\phi(0) \gg \phi_+$, the correspondingly large field displacement inside the (proto)-bubble makes it easier for a bubble to escape from the star.

The key point is that, for what concerns the possibility of a bubble of the true vacuum escaping from the star, one only needs to focus on a "sub-bubble" with a field displacement $\Delta\phi_{\rm sub} = \phi_+ - \phi_- \approx 2f$. The energy density of such a sub-bubble is simply $\epsilon_{\rm sub} \sim \Lambda_{\rm R}^4$, while its tension scales as

$$\sigma_{\rm sub}(R_{\rm T}) \sim \sqrt{\Delta\phi(0)f}\,\Lambda_{\rm R}^2. \qquad (64)$$

---

[12]This requirement does not depend on $\sigma'$ being constant, and the condition on $R_{\rm T}$ in Eq. (62) holds in general with $\sigma' \to \sigma'(R_{\rm T})$, under our approximation that $\sigma'$ turns on at $R_{\rm T}$.

The latter is enhanced by a factor $(\Delta\phi(0)/\Delta\phi_{\text{sub}})^{1/2}$ with respect to the naive expectation, due to the higher potential energy difference of the large (proto-)bubble that contains the sub-bubble, $|\langle\Delta V\rangle| \sim \Delta\phi(0)\Lambda_{\text{R}}^4/f$. This simple estimate holds as well if we assume that the in-density minimum is reached, i.e. $\phi(0) = (\phi_+)_n$.

Such an enhancement of the tension facilitates the escape of the sub-bubble, since it decreases the contracting force associated with $\sigma'$ in Eq. (37). In particular, we now have $\sigma'_{\text{sub}} \sim [\sigma(R_{\text{S}})-\sigma_{\text{sub}}(R_{\text{T}})]/\Delta R_{\text{T}}$, which is smaller than when $\phi(0) \sim \phi_+$, see Eqs. (38), (39); in fact it could even be negative. Notice that instead the force associated with the surface tension of the wall at the transition radius, $2\sigma_{\text{sub}}(R_{\text{T}})/R_{\text{T}}$, remains constant, since $R_{\text{T}} \sim \sqrt{\Delta\phi(0)/f}\,\mu^{-1}$. Therefore, the net result is that it is much easier for the escape condition Eq. (40) to be satisfied. The larger (proto-)bubble supporting the sub-bubble helps the latter permeate through the entire star. The proper condition that determines if the sub-bubble of true vacuum expands throughout the whole universe is then

$$R_{\text{S}} \gtrsim \frac{2\sigma(R_{\text{S}})}{\epsilon}\,. \tag{65}$$

We have explicitly verified this result via our numerical simulations. For a bubble connecting shallow minima, $\delta^2 \ll 1$, this condition translates into

$$R_{\text{S}} \gtrsim \frac{f}{\Lambda_{\text{R}}^2}\,, \qquad \text{(sub-bubble; shallow)} \tag{66}$$

a requirement that is automatically satisfied given that $R_{\text{S}} > R_{\text{T}}$. For a bubble connecting deep minima, $\delta^2 \approx 1$, we find instead

$$R_{\text{S}} \gtrsim \frac{f}{\Lambda_{\text{R}}^2}\frac{1}{\sqrt{1-\delta^2}}\,. \qquad \text{(sub-bubble; deep)} \tag{67}$$

This is similar to the escape condition for a deep bubble, Eq. (42), yet on $R_{\text{S}}$ instead of $\Delta R_{\text{T}}$.

# E  Sudden approximation

We have been assuming that the bubble, during either its formation or expansion through the star, is always found in a nearly-static ($\dot{R} = 0$) equilibrium position, with its radius evolving slowly only because $R_{\text{T}} = R_{\text{T}}(\bar{t})$ does, as the star is being formed. Only at the point where equilibrium is lost, $\ddot{R} > 0$ and the bubble is free to gain kinetic energy. This was justified in Sec. 3 on the basis that the characteristic reaction time of the scalar field, $\mu^{-1}$, is much shorter than the evolution time of the star $T_{\text{s}}$. In this section we wish to comment on the opposite situation, where $\mu T_{\text{s}} \ll 1$.

In this limit, the star is formed instantaneously, with a large region $r < R_{\text{T}}$ where the in-density potential allows for the scalar field to start classically rolling. If such a region was of infinite extent, i.e. if the system was spatially homogeneous, the field would roll, accelerate, and finally oscillate around the true minimum. However, in a finite-size system, one needs to crucially take into account the contribution of the spatial gradient to the energy of the field configuration. Indeed, $\phi$ moves in an effective potential $V(\phi) + \frac{1}{2}\phi'^2$ that becomes large towards the transition region, where the field must return to its vacuum value $\phi_-$. Therefore, the sudden formation of the star and the corresponding gain of kinetic energy $\frac{1}{2}\dot{\phi}^2$ does not automatically imply that a first order phase transition will proceed via the escape of a scalar bubble from the dense system. As a matter of in fact, the situation is not much different that in the quasi-static case, as we now explain.

Concerning the formation of the bubble, the main difference with respect to our discussion in Sec. 4.1 can be phrased in terms of the maximal value that $\Delta\phi(0) = \phi(0) - \phi_-$, the field displacement at the center of the star, can take. Indeed, because of the kinetic energy the field acquires by rolling down the in-medium potential, $\Delta\phi(0)$ will generically be larger than what found in Eq. (22) for the same $R_\mathrm{T}$, yet oscillating in time. Accordingly, the whole scalar profile will necessarily oscillate in time as well. Then, if the size of star, specifically $R_\mathrm{T}$, is still not large enough for $\phi(0)$ to reach $\phi_+$, the field value corresponding to the true minimum of the scalar potential in vacuum, then such an oscillating scalar profile remains trapped within the star, in a sort of oscillon that, even after eventually losing its kinetic energy,[13] remains as a confined static bubble (see e.g. [60] for a recent discussion of such type of field configurations in vacuum).

Otherwise, if $\Delta\phi(0) \gtrsim 2f$, then whether the scalar bubble remains confined to the dense region or escapes to infinity follows from the same analysis as in Sec. 4.2, yet with the properties of the bubble, i.e. the potential energy difference between the two sides of the bubble wall and the tension, now oscillating in time.

We stress again that the main difference between the quasi-static and sudden scenarios concerns the value of $R_\mathrm{T}$ for which a given field displacement is attained. Another way to interpret this fact is to compare, for the same value of $R_\mathrm{T}$, the dynamics of the bubble wall between the two scenarios. Because of the larger field displacement in the sudden case, the maximum values of $\epsilon(t)$ and $\sigma(R, t)$ will both be larger, while $\sigma'(R, t)$ will be smaller, than in the quasi-static case. This situation resembles the quasi-static evolution of a bubble in the limit that $n \gg n_c$, discussed in App. D. Therefore, we could similarly conclude that in the sudden approximation and for $R_\mathrm{T} \gg \mu^{-1}$, the condition that determines if the bubble expands indefinitely is

$$R_\mathrm{S} \gtrsim \frac{2\sigma(R_\mathrm{S})}{\epsilon} \, . \tag{68}$$

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
