# Peer review of "Density Induced Vacuum Instability"

_SciPost Physics_

## Round 1 · Referee Report · Anonymous (Referee 1) · 2022-2-18

Report

The authors study false vacuum decay catalysed by finite-density effects. A simple model of the real scalar field is considered. The scalar field potential is chosen to have two non-degenerate minima corresponding to the false and true vacua. The scalar field is assumed to be coupled to other fields whose ground state changes the parameters of the potential, thus reducing or removing completely the barrier between the two vacua. The ground state assumes the compact profile which is a prototype of an ordinary star, a neutron star or an exotic compact object. The authors study classical nucleation of a true vacuum bubble inside such an object and its subsequent dynamics. They also comment on the possibility of quantum tunneling in the case when the barrier is not completely removed.

The topic of vacuum decay in field theory is important nowadays both from the theoretical and phenomenological points of view. The current manuscript provides a well-motivated and detailed contribution to this topic. Nevertheless, I have a question and a few small points for the authors to consider, before I can recommend it for publication.

1) The authors conclude that (baryon) stars can catalyse vacuum decay due to high densities of nuclei in their cores, if the potential of the decaying scalar field is chosen appropriately. However, at sufficiently early times the Universe was filled with much denser (and hotter) radiation than any star can provide today. Thus, any (classical) vacuum transition, which could be triggered by stars, would be triggered during the hot Big Bang. Therefore, an explanation would be useful as of if and why stars can still serve as bubble nucleation cites at later epochs.

Let me now make some comments concerning Sec. 4.4 where tunneling is discussed.

2) After Eq. (4.29), the authors say: "...the bounce action at finite density can only be sufficiently small in absolute terms if $(f/\Lambda_B)^4= 1/\lambda$ is small...". It is not clear what "sufficiently" means here. The decay probability can be exponentially suppressed (if $1/\lambda$ is large) but still important, if the time scale (the age of the Universe, for example) is large enough.

3) In the same paragraph, the authors say: "...seeded nucleation of bubbles of the true ground state would generically not take place during the formation of the star...". In my opinion, vacuum decay catalysed by short but violent processes, such as the star explosion or the black hole formation, is an interesting open question, and a general statement here might be premature.

4) In the next sentence ("On the contrary, one would expect...") the authors imply that the decay probability accumulated during the lifetime of the star, should be of order one. This condition is relaxed if one considers a system of many stars, each serving as the bubble nucleation cite.

Further minor comments include:

5) Sec. 4.4 mentions some papers studying false vacuum decay catalysed by black holes (Refs. [39-43]). The authors might also be interested in the recent papers 2105.09331, 2111.08017 on this topic.

6) In Sec. 5, in the paragraph following Eq. (5.1), the authors conclude that large stars are needed for the phase transition seeded by them to fit the cosmological data. It might be interesting in this respect that first stars in the Universe (so-called pop III stars) could be very large and massive; see, e.g., 1501.01630.

7) First-order phase transitions are generally accompanied by production of gravitational waves. In view of the current interest to this topic, it would be nice if the authors could comment on the prospects of detection of gravitational waves produced by vacuum transitions studied in their work.

  • validity: -
  • significance: -
  • originality: -
  • clarity: -
  • formatting: -
  • grammar: -

Author:  Konstantin Springmann  on 2022-06-10  [id 2574]

(in reply to Report 1 on 2022-02-18)

We thank the referee for the insightful comments and interesting questions on our manuscript. Below we address in detail each of the points raised and, where appropriate, list the corresponding changes we have made.

1) Classical vacuum transitions in the early universe

In our paper we consider a simple potential with two non-degenerate minima, the higher of which gets destabilised by baryonic-density effects thus triggering a phase transition. We completely agree with the referee that such a phase transition could a priori happen as well in the early universe, in particular due to finite temperature effects. This would depend on the temperature necessary to destabilize the metastable minimum, e.g. $T \sim 100 \, \text{MeV}$ at the quark-hadron transition would roughly correspond to the densities found in neutron stars $n \sim (100 \, \text{MeV})^3$, or the higher $T \sim 100 \, \text{GeV}$ if we consider a barrier that depends on the Higgs VEV. However, it could well be that such high temperatures were never reached in the early universe (if reheating temperatures were lower). In this respect, not much is known about the hot history of the universe for temperature above BBN ($T \sim \text{MeV}$), where baryonic densities were very low $\rho_{\tiny{\text{B}}} \sim 10^{-10} \, \text{MeV}^3$ (see e.g. Hook 2017). For this reason, we chose to be agnostic about the fate of the metastable vacuum in the early universe. Besides, we would like to point out as well that one of our motivations for considering such type of potentials is the relaxion, were the two minima we have focused on are embedded in a landscape containing many non-degenerate minima. It has been shown in Graham 2015 and Choi 2016 that, due to the overall flatness of the relaxion potential, even if reheating temperatures were high enough as for the barriers between minima to disappear, the universe would have cooled so fast that the relaxion would not have rolled many minima down the potential. This means that the relaxion would have eventually landed in some minimum that is susceptible to being destabilised by stellar compact objects.

We have added a couple of sentences elaborating on this point at the end of section 2.

2) Small bounce action for vacuum decay

The referee is right in that to assess if the decay probability is large enough for the in-density decay to take place, the bounce action $\textit{and}$ the time $\mathcal{T}$ and volume $\mathcal{V}$ are relevant. We did have in mind rough estimates for what the latter are, which we present in the following. Still, the decay rate is exponentially sensitive to the value of $(f/\Lambda_B)^4 = 1/\lambda$, so the message we tried to convey is that a strong coupling $\sqrt{\lambda} = O(1)$ is generically required, as we now show: The tunneling rate per volume is given by \begin{equation} \Gamma/\mathcal{V}=Ae^{-S_{\tiny{\text{B}}}}, \end{equation} where $S_{\text{B}} \sim 27 \sqrt{3} \pi^2 (1-\zeta)^2 (1-\delta^2)^{-3} \lambda^{-1}$ for a deep minimum and $A \sim f^4$, with $\zeta = \zeta(n)$ encoding the effect of a non-zero density, $\zeta(0) = 0$ and $\zeta(n>0) \neq 0$. The decay probability becomes order one, i.e. $\Gamma \mathcal{T} = O(1)$ for \begin{equation} \frac{\lambda (1-\delta^2)^3}{(1-\zeta)^2} \sim \frac{27 \sqrt{3} \pi^2}{\log(f^4 \mathcal{T} \mathcal{V})} \, . \end{equation} The volume of all the neutron stars in the observable universe can be roughly estimated as $V_{\text{NS}} \sim R_{\text{NS}}^3 N_{\text{NS/G}} N_{\text{G}} \sim 10^{29} \, \text{m}^3 $, where $R_{\text{NS}} \sim 10 \, \text{km}$ is the typical radius of a neutron star, $N_{\text{NS/G}}\sim 10^9$ the number of neutron stars per galaxy (like ours) and $N_{\text{G}} = 10^{11}$ the number of galaxies. As the relevant time scale one can just take the age of the universe, since star formation happened relatively early, $\tau\sim 1/H_0 \sim 10^{17} \, \text{s}$. One then finds for $f < M_{\tiny\text{Pl}}$, \begin{equation} \frac{\lambda (1-\delta^2)^3}{(1-\zeta)^2} \gtrsim 1.0 \,. \end{equation} Note that such a value barely changes if we consider instead the volume of the whole universe, $V_{U} \sim 1/H_0^3 \sim 10^{78} \, \text{m}^3$. Therefore, the the change in $\zeta$ is the important parameter for the decay of the metastable minimum taking place in density and not in vacuum, as we explained in the text. In the best case scenario where the metastable minimum is not very deep, e.g. $\delta \approx 0.8$, considering a mild reduction of the potential barrier in density, $\zeta \approx 0.5 < \delta^2$, is enough to catalyse the decay of a minimum that is essentially stable in vacuum. From the inequality above this means that $\sqrt{\lambda} \gtrsim 2.9$, which is certainly not a weak coupling. Deeper metastable minima require even larger coupling.

We have rewritten the sentence below Eq. (45) and added some further details to make this point more clear.

3) Vacuum decay catalysed by short but violent processes

All we wanted to say with the statement that ``...seeded nucleation of bubbles of the true ground state would generically not take place during the formation of the star...'' is that the time scales of vacuum decay ($T_B \equiv 1/\Gamma$) and star formation ($T_S$) are unlikely to coincide. This is in fact pointed out in the sentence immediately after, where we also state that the lifetime of the star (i.e.~the time since its formation, $\mathcal{T} \sim 1/H_0$) could be much larger. In addition, let us note that the violent process of a star explosion or the formation of a black hole would be a classical effect not directly related to vacuum decay, which is the focus of this section.

4) System of many stars

We completely agree with the referee. We already took this fact into account since to derive the decay rate we considered the total volume in stars, as explained in detail in point 2 above.

5) Vacuum decay catalysed by black holes

We thank the referee for bringing these recent references to our attention. We have now included the corresponding citations.

6) Pop III stars

We thank the referee for pointing out the possibility that Pop III stars could have acted as nucleation seeds for phase transitions consistent with cosmological data. We have added a sentence and provided further details on the properties of this hypothetical population of stars.

7) Gravitational waves

We agree with the referee that it would be very interesting to study the potential gravitational wave signals associated with the phase transitions we have studied. Along with their detailed cosmological dynamics, this is something we are currently investigating. We are stating this now in section 5.

---

## Round 2 · Referee Report · Anonymous (Referee 2) · 2022-7-10

Report

The paper explores the consequences of finite density effects for vacuum instability. It is argued that high-density systems such as stars might destabilize a metastable minimum (basically by removing the potential barrier) and allow for the formations of bubbles of the true minimum which, under some conditions, might escape the dense system and extend to infinity. The authors mainly focus on the case where the scalar field can classically move to the true minimum of the potential, although they also briefly analyze the case where the barrier has not completely disappeared and the transition happens via quantum tunnelling. These considerations apply to a general class of scalar potentials, featuring a density dependent potential barrier. Paradigmatic examples are relaxion potentials, which are set by the QCD quark condensate or by the Higgs VEV, both acquiring a non-trivial dependence from finite density. The phenomenological consequences of these density-induced vacuum transitions are intriguing: late phase transitions at stars formations which change the vacuum energy and might leave an imprint in cosmological observations.

The paper is well structured and clearly written. Moreover, I find the results original and relevant, and therefore I recommend that the present study is published in SciPost Physics, after the following minor points are addressed:

  1. It would be good to clarify better the role of density effects on the rolling term in the potential in eq. 1. Is the assumption of neglecting the density dependence in Lambda_R motivated in some explicit realizations ?

  2. sect. 4.4: it seems to me that the author never consider the case in which the minimum is deep, but not in the thin-wall regime. Is this because the authors have in mind a specific class of models?

  3. Related to the previous question, are there potential consequences of finite-density effects on the SM vacuum decay rate?

  • validity: high
  • significance: high
  • originality: top
  • clarity: high
  • formatting: excellent
  • grammar: excellent

Author:  Konstantin Springmann  on 2022-09-22  [id 2839]

(in reply to Report 1 on 2022-07-10)

We thank the referee for the insightful comments and interesting questions on our manuscript. Below we address in detail each of the points raised.

1) Density dependent rolling scale

In this work, neglecting the density dependence of the rolling term is motivated by the explicit realization of relaxion models. In these models, although the rolling term is also slightly density dependent, the dominant effect due to SM matter density is the change in the height of the barriers. However, we agree with the referee that a density-dependent rolling scale is indeed interesting. In fact, in a subsequent work Balkin:2021, we investigated a technicolored relaxion where the rolling scale is actually increased due to a background electro-magnetic field, and similar conclusions as for a matter density dependent backreaction scales hold.

2) Barrier height vs. bubble thickness

The case of a deep minimum following our potential in Eq. 1 is by construction the case where the barrier height is much larger than the energy difference of the two vacua. This means by definition that we are in the thin wall case. In that sense, having a deep minimum is equivalent to having a thin-wall bubble. Of course, one could have parameters such that the potential is neither deep nor shallow, in which case the thin wall approximation would not hold.

3) Stability of the SM vacuum

The destabilization of the Higgs potential typically occurs at high scales, i.e. $\Lambda_{UV}\sim10^{12} \,\text{GeV}$, see Degrassi:2012ry. The linear coupling of the Higgs to SM density effectively adds a term to the Higgs potential of the order of $\Delta V \sim\rho H$. However, this linear term cannot compete with a quartic in the potential $\lambda H^4$ at $\left\langle H\right\rangle\sim 10^{12} \text{GeV}$ even at the highest known densities in compact SM objects $\rho \lesssim \text{GeV}^3$ .

Anonymous on 2022-11-25  [id 3073]

(in reply to Konstantin Springmann on 2022-09-22 [id 2839])

We thank the referee again for the insightful comments and interesting questions.

We now added minor changes, as suggested.

1) Concerning the density dependence of the rolling scale, we would like to point the reader to the paragraph below Eq. 5, where we give explicit models where the dominant density dependence is in the backreaction scale. Below Eq. 6 we added a sentence where we refer to a model where the dominant effect is due to a background dependent rolling scale. Note that there the role of the background is played by electromagnetic fields.

2) We added a sentence in section 4.4 which clarifies that with our potential the thin wall approximation always holds for deep minima.

3) We added a sentence above Eq. 16 which points out the relevant scales at which our effect is important. Since the scales required to destabilize the SM vacuum are much higher, it can never be destabilized by the mechanism described.

---

## Round 2 · Referee Report · Anonymous (Referee 1) · 2022-8-15

I am satisfied with the changes made by the authors. In my opinion, the paper can be published in SciPost Physics.

---

## Editorial Decision

accepted_in_target_journal